# Nanoscale kinetic segregation of TCR and CD45 in engaged microvilli facilitates early T cell activation

Yair Razvag[1,2], Yair Neve-Oz[1], Julia Sajman[1], Meital Reches[2] & Eilon Sherman [1]

T cells have a central function in mounting immune responses. However, mechanisms of their early activation by cognate antigens remain incompletely understood. Here we use live-cell multi-colour single-molecule localization microscopy to study the dynamic separation between TCRs and CD45 glycoprotein phosphatases in early cell contacts under TCR-activating and non-activating conditions. Using atomic force microscopy, we identify these cell contacts with engaged microvilli and characterize their morphology, rigidity and dynamics. Physical modelling and simulations of the imaged cell interfaces quantitatively capture the TCR–CD45 separation. Surprisingly, TCR phosphorylation negatively correlates with TCR–CD45 separation. These data support a refined kinetic-segregation model. First, kinetic-segregation occurs within seconds from TCR activation in engaged microvilli. Second, TCRs should be segregated, yet not removed too far, from CD45 for their optimal and localized activation within clusters. Our combined imaging and computational approach prove an important tool in the study of dynamic protein organization in cell interfaces.

[1] Racah Institute of Physics, The Hebrew University, Jerusalem 91904, Israel. [2] Institute of Chemistry, The Hebrew University, Jerusalem 91904, Israel. Correspondence and requests for materials should be addressed to E.S. (email: eilonsher@gmail.com)

The physical interaction of T cells and antigen-presenting cells (APC) enables the recognition of cognate foreign antigens and the subsequent mounting of an appropriate T-cell-mediated immune response. The specific and sensitive recognition of foreign antigens is performed by the T-cell antigen receptor (TCR), which then initiates a signalling cascade towards multiple effector functions[1]. The TCR signal is carefully regulated, since its over reactivity may cause auto-immunity and graft rejection, while TCR reactivity that is too weak may cause anergy. In spite of the importance of TCR activation to human health, its detailed underlying mechanisms have not been fully resolved. Diffraction limited microscopy has shown that the TCR and downstream effectors form pronounced clusters[2,3] and that TCR triggering and $Ca^{++}$ influx occur within seconds of first engagement of TCRs with cognate antigens[4,5]. Results from super resolution imaging of these clusters have shown that the TCR and related signalling molecules come together in nanoclusters[6,7] that can form dynamic and heterogeneous functional nanoscale patterns[7,8]. Importantly, unexplained localized and synchronized activation of TCRs within larger TCR clusters has been observed[9,10].

Another type of molecular patterning at the immune synapse (IS) involves the physical separation of engaged TCRs from bulky glycoproteins in tight contacts[11]. This separation has been proposed to remove continuous phosphatase quenching of basal TCR signals by proximal CD45 glycoproteins and allow the propagation of the TCR signal downstream[12]. However, this separation, called kinetic segregation (KS), has been mostly shown in mature contacts between T cells and APCs[11] that take minutes to develop. Thus, the observed KS in such contacts seems too late to influence early T-cell activation. Moreover, Chang et al.[13] resolved KS in early contacts of T cells with activating surfaces. Still, multiple critical issues remain unresolved, since KS within these contacts occurs fast (within seconds) and at the nanoscale and thus, cannot be fully resolved by diffraction limited microscopy[14]. First, the nature of the physical contacts shown by Chang et al. remains unclear. Second, the nanoscale spatio-temporal relation of TCR clusters and KS within these contacts and during cell spreading has not been resolved[15]. Third, the relation of KS to TCR nano-clusters and micro-clusters, and the localized activation of TCRs within clusters[9] have not been studied. Specifically, the dual role of CD45 in Lck activation and in dephosphorylating ITAMs on intracellular TCR chains requires its fine-tuned positioning in respect to TCR clusters and esp. to phosphorylated TCRs (pTCR). Last, physical models of the KS predict a critical nanoscale depletion distance between the TCR and CD45[16], which cannot be resolved using diffraction limited microscopy. Such a depletion, if exists, is a direct evidence for the mechanical forces that act by the PM, the TCR and its ligands, and the related glycoproteins (primarily, CD45). Measuring this distance could become invaluable in understanding the mechanics of the membrane and molecules that facilitate TCR triggering[15,16]. Arguably, resolving of these open issues is required in order to establish a unified physical model of early T-cell activation by the TCR[15].

Here, we study the KS of the TCR from CD45 at the PM of live T cells using single-molecule localization microscopy (SMLM). For that, we establish a two-colour approach that combines photoactivated localization microscopy (PALM)[17] and direct stochastic optical reconstruction microscopy (dSTORM)[18]. SMLM imaging results and second-order statistics show a physical separation between these molecules in early forming contacts under a range of TCR-stimulating and non-stimulating conditions. This separation grows over time for TCR-stimulating conditions, yet is much reduced under non-stimulating conditions. Atomic force microscopy (AFM) and SMLM further serve

to identify the early contacts with touching microvilli and to characterize their morphology, rigidity and dynamics. Surprisingly, TCR triggering, as indicated by pTCRζ, negatively correlates with the physical separation between TCRs and CD45. To test our understanding of the KS and its underlying mechanisms, we employ a physical model of the IS and dynamic simulations of the cell interface in molecular detail. Our simulations capture the KS and its dynamics under the range of experimental conditions. Based on our results, we propose a refined view of the KS model where it occurs within seconds from TCR activation in engaged microvilli and that TCRs should be segregated, yet not removed too far, from CD45 for their optimal activation within clusters, thus shedding new light on this critical process of T-cell activation.

## Results

**Two-colour SMLM of live cells**. To study the dynamic nanoscale organization of the TCR and CD45, we first developed a two-colour SMLM approach that combines PALM and dSTORM for imaging of proteins at the PM of live T cells. For dSTORM we used immunofluorescence of the extracellular parts of membrane proteins without cell fixation. Direct-STORM uses a cocktail of oxygen scavengers and thiols to optimize photoswitching (see Methods, Sample preparation). These reagents are largely incompatible with the imaging of fluorescent proteins. Nevertheless, we found that Dronpa, a green photoactivatable fluorescent protein (PAFP)[19], can still be used under those conditions for PALM (Fig. 1a). Thus, our SMLM imaging approach included the labelling of TCRζ with Dronpa (TCRζ–Dronpa) for PALM. For dSTORM, we used immunofluorescence staining with a primary αhuman-CD45 and a secondary αmouse antibody labelled with Alexa647 in dSTORM imaging buffer (see Methods, Sample preparation). Using this approach, we were able to image and localize these molecules with a spatial resolution down to 20–30 nm (Supplementary Fig. 1A). We verified the registration of these channels (Supplementary Fig. 1B, C), and that no cross-talk existed between the two imaging channels (Supplementary Fig. 1D).

**Physical separation of TCR and CD45 in early forming clusters**. The KS model suggests that the TCR and CD45 dynamically segregate during cell activation. Indeed, our imaging of the IS (Fig. 1a) of fixed cells, captured at different stages of synapse formation, shows the existence of multiple foci of CD45 rings around TCR clusters in an early spreading cell (Fig. 1a, top cell in middle image), while these foci coalesce into a single synapse (with a single CD45 ring) for a well spread cell (Fig. 1a, bottom cell in middle image)[20]. This result indicates the need for fast SMLM imaging of the IS upon cell spreading. For such imaging, we accumulated SMLM movies at a frame rate of 50–100 fps and an effective temporal resolution of 2.5 s for constructing SMLM images. This temporal resolution was sufficient to capture the dynamics of synapse formation (Fig. 1b and Supplementary Movies 1, 2). Imaging of cell spreading on anti-CD3- and anti-CD45-coated coverslips enabled us to observe the correlation between the TCR and CD45 during synapse formation under these conditions. The dynamic correlation was characterized by bivariate pair-correlation functions (BPCF) (Fig. 1c and Supplementary Fig. 3A, B). Through this analysis, non-interacting species result in a flat $g_{12} = 1$ function, referred to as the non-interaction (NI) model[21]. This statistics can further report on the interaction between binding proteins, when compared to BPCF due to the random labelling (RL) model[21]. Through this model, the labels of detected proteins are randomly relabelled while keeping their positions. The maximal and minimal values of

BPCFs due to 19 Monte-Carlo simulations can set the 95% confidence interval of the RL model[22]. The BPCF of strongly interacting proteins tends to follow the confidence interval of the RL model[21]. Here, however, we used this statistics to identify the segregation (i.e. negative correlation) between the TCR and CD45, resulting in the departure of the BPCF from the RL model confidence interval.

These bivariate functions were standardized according to the RL model (SBPCF$_{RL}$) for comparison between different time points and different stimulating conditions (see Fig. 1d, Supplementary Fig. 3C, D and Methods, Statistics)[23]. Standardized bivariate pair-correlation functions (SBPCF) curves due to the RL model result in a flat curve at 0. Negative deviation of the SBPCF curves from 0 indicates the co-occurrence of partial attraction (i.e.

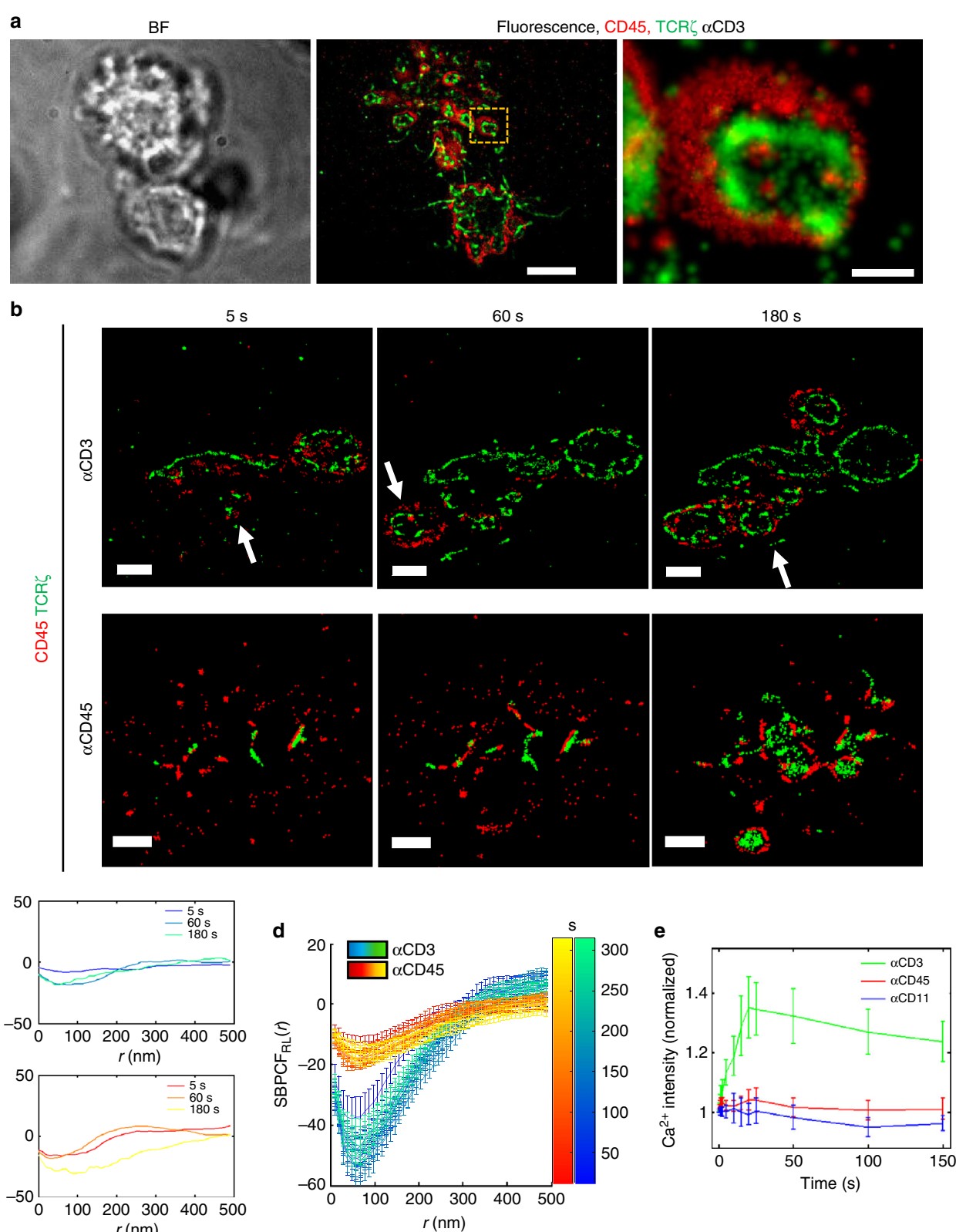

co-localization) and partial repulsion (i.e. segregation) between the interacting species. The $SBPCF_{RL}$ clearly shows a specific distance of segregation for TCR and CD45 up to ~350 nm. Surprisingly, this phenomenon significantly increased with time, reaching a peak after ~150 s from the start of cell spreading and then reduced back to starting values after ~300 s (Fig. 1d; $p \ll$ 0.001). Although the segregation of the TCR and CD45 occurred under both TCR-stimulating and non-stimulating conditions, it had more significant negative SBPCF values under stimulating conditions ($p \ll 0.001$). Supplementary Figure 3E,F further shows the substantial dynamic evolution of the KS process on anti-CD3-coated coverslips.

To validate our results, we conducted both negative and positive control experiments on T cells that spread on αCD3-coated coverslips and were fixed after 4 min. A negative control showed that two unrelated proteins (Syntaxin 1A and CD45) did not colocalize, and followed the NI model up to ~150 nm (Supplemental Fig. 2A, B), with deviations at longer scales that likely arise from PM patterning[21,23]. In contrast, PALM imaging of TCRζ–Dronpa and dSTORM imaging using αTCRα/β staining resulted in high colocalization and SBPCF statistics that closely followed the RL model (Supplementary Fig. 2C, D). Reversal of the imaging channels of TCR and CD45 captured again the segregation between these molecules (Supplementary Fig. 2E), with SBPCF statistics showing a similar separation between these molecules (Supplementary Fig. 2F, H).

Calcium imaging experiments showed a robust $Ca^{++}$ flux for cells spreading on anti-CD3 rather than on anti-CD45-coated coverslips (Fig. 1e). Thus, the protein segregation was more efficient under TCR-stimulating conditions in comparison to non-stimulation conditions.

After observing numerous cases of live cells encountering anti-CD3-coated coverslips, we found a characteristic pattern of spreading. At first, a medium-sized synapse (of ~4 μm) formed between the cell and the surface (big elongated area in Fig. 1b). The point patterns of TCR and CD45 in this area were typically separated and stable. However, once the cell committed to spreading, new small and remote regions formed and gradually spread till fusing together with the main region and with each other (white arrows in Fig. 1b).

**Formation of a growing depletion zone between TCR and CD45.** We were specifically interested in the newly forming contact regions described above. We could analyse their point patterns starting at the moment they formed and with a relatively low background, since these regions were isolated from other regions in the cell footprint (Fig. 2a, and Supplementary Fig. 4A and Supplementary Movies 3, 4). When zooming onto small scales, the relatively fast decay of Dronpa fluorescence became detrimental[24], as the amount of data available limited our SBPCF statistics. To improve the visibility of our SMLM imaging, we considered also photobleached TCRζ–Dronpa molecules and assumed that their binding to immobilized anti-CD3 antibodies on the coverslip surface restricted their diffusion. With these assumptions, we utilized Kalman filtering, which provides the ability to retain the position of the molecules for a defined time

during the analyses. The retention time was determined according to a gain parameter (see Supplementary Fig. 4B and Methods, Analysis and presentation). Kalman filtering was not necessary for Alexa647, due to its robust emission for dSTORM[25].

To develop a quantitative analysis of the segregation of CD45 from TCR molecules, we simulated a model (Supplementary Fig. 4C), where a ring representing the cell region containing CD45 molecules expands from an inner disk containing the TCR molecules. Both the disk and the ring grew linearly in size, but at different rates (Supplementary Fig. 4E). At $t = 0$, the simulated CD45 and TCR molecules positively mixed. As expected, the $g_{12}$ function did not cross the NI model limits in the range of 0–500 nm (Supplementary Fig. 4D). As the mutual pattern evolved, CD45 molecules segregated towards the periphery of the region and a 'depletion zone', a zone with no TCR or CD45 molecules, started to form (white arrows in Supplementary Fig. 4C). The $g_{12}$ function captures the creation and expansion of the depletion zone by shifting its crossing of the curve due to the NI model to longer length-scales. Meaning, the correlation between CD45 and TCR molecules still existed, but not at short mutual distances where the molecules segregated. The depletion zone size can be characterized by the distance in which the correlation returns to be positive again (i.e. where $g_{12}$ crosses the NI model; black arrows in Supplementary Fig. 4D).

Returning to the experimental results, we first quantified the cluster sizes of the TCR and of CD45 as the cells spread on either αCD3-, αCD11a- or αCD45-coated coverslips. The maximal cluster sizes were quantified using the univariate PCF of the TCR and of CD45, according to the largest scale showing significant clustering in these statistics (Methods)[4]. We observed that the maximal size of TCR clustering remained relatively stable during the spreading time under all conditions (Fig. 2c), while CD45 clusters reduced their maximal size over time in a similar fashion under all conditions (Fig. 2d). Surprisingly, under all conditions, a depletion zone formed and grew over time (Fig. 2e). For αCD3-coated coverslips, it reached a plateau at about 230 nm after 25 s. Still, the segregation dynamics was significantly faster on anti-CD3- and anti-CD11a-coated coverslips vs. on anti-CD45-coated coverslips (Fig. 2e and Supplementary Fig. 4F; $p \ll 0.001$).

So far, we have used staining of CD45 with a primary and a secondary antibody for dSTORM. To minimize the effect of the antibodies size on the apparent segregation of the molecules, we conducted similar live-cell imaging on cells stained with a primary αCD45, directly conjugated to Alexa647 (Supplementary Fig. 5A). Our imaging resulted in similar KS and similar depletion distance and dynamics (Supplementary Fig. 5A, B) of TCR and CD45 at the PM of cells.

We next studied whether KS occurred for a second T-cell line of a different lineage. For that, we conducted live cell PALM/dSTORM imaging of TCRζ–Dronpa and CD45 at the PM of J76 $CD8^{+}$ cells (see Methods, Cell lines) on αCD3-coated coverslips. Our imaging showed distinct KS between TCR and CD45 also in these cells, with comparable depletion distance and dynamics to our results for Jurkat E6.1 cells (Supplementary Fig. 5C, D).

Since we detected differences in the depletion distance under TCR-stimulating and non-stimulating conditions (Fig. 2a, b, e), we tested the effect of TCR stimulating levels on the KS between

**Fig. 1** Physical separation of TCR and CD45 in early forming clusters. **a** (left) Bright-field and (middle) two-colour SMLM, PALM combined dSTORM of Jurkat cells expressing TCRζ–Dronpa (green) and CD45 immunostained with Alexa647 (red) spread on an anti-CD3 coated coverslip 4 min before fixation. Scale bar 5 μm. (right) Zoom of the marked area in **a** (middle). Scale bar 200 nm. **b** Two-colour SMLM, PALM combined dSTORM of live Jurkat cells expressing TCRζ–Dronpa (green) and CD45 immunostained with Alexa647 (red) spread on an anti-CD3 (left panel) and anti-CD45 (right panel) modified coverslips at different time samples of 5, 60 and 180 s after the cell started to spread. White arrows indicates new forming regions. Scale bar 2 μm. **c** Plots show the PCF between the TCR molecules and CD45 molecules at the corresponding time positions. **d** Time evolving bivariate PCFs normalized by RL model (SBPCF) between TCR molecules and CD45 molecules on an anti-CD3 (winter) and anti-CD45 (autumn) modified coverslips. **e** Calcium imaging experiment shows the dynamic $Ca^{++}$ flux (as an activation state mark) followed Jurkat E6.1 cells spreading over various coated coverslips

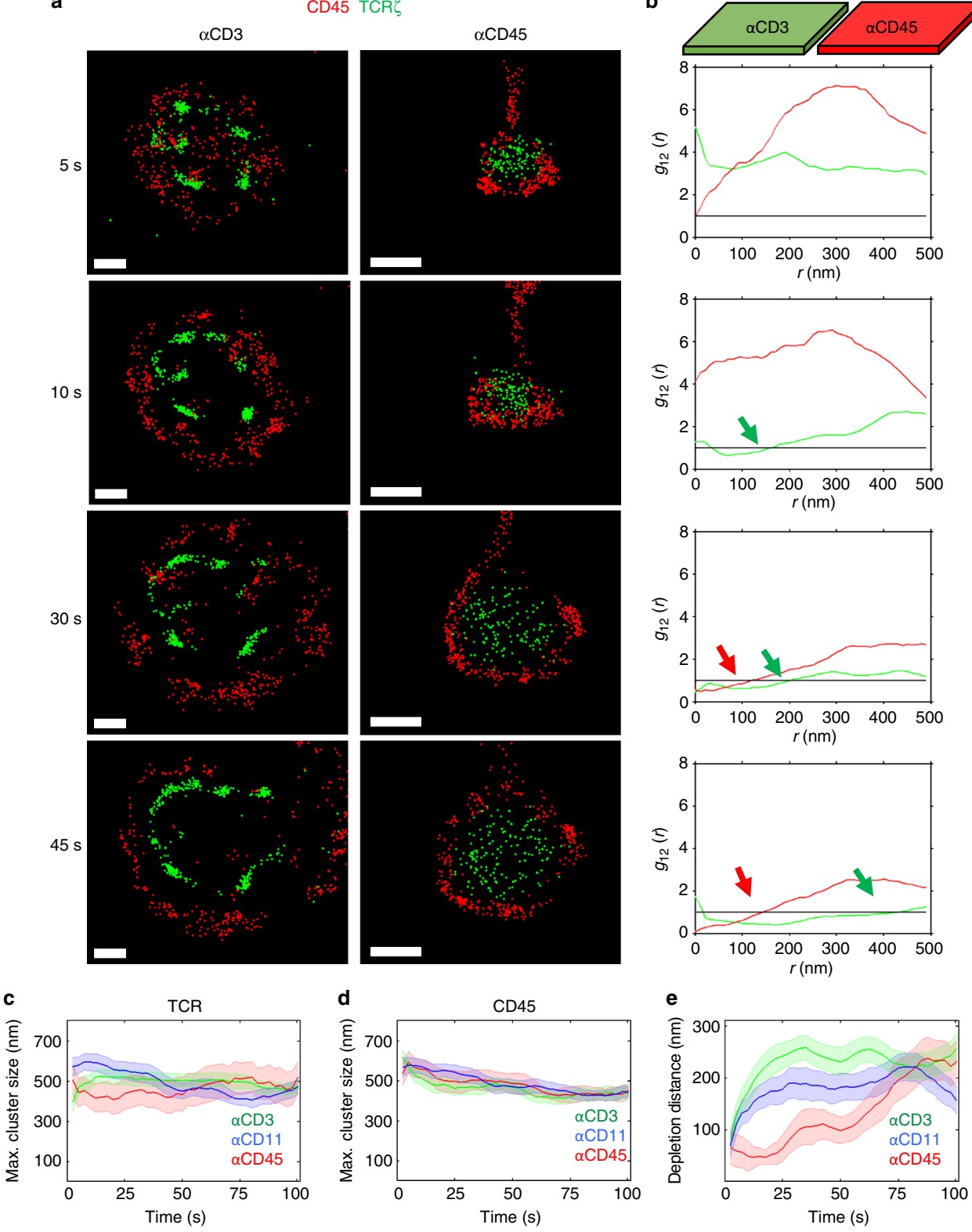

**Fig. 2** The evolution of a depletion zone and its dynamics under multiple conditions. **a** Zooming to new forming regions. Time sequence of the synapse formed at distinct areas in between Jurkat cells expressing CD3ζ-Dronpa (green) and immunostained for CD45 with Alexa647 (red) and modified coverslips, anti-CD3 (left) and anti-CD45 (right) at different time samples of 5, 10, 30 and 45 s after the cell started to spread. Scale bar 0.5 μm. **b** The PCF between TCR and CD45 molecules captured from the areas shown in **a** of cell spread over anti-CD3 (green lines) and anti-CD45 (red lines) modified coverslips. Arrows point to the crossing of the PCF with the value of one. The position of where the PCF is crossing the value of one define the: (1) maximal TCR cluster size **c**, (2) the maximal CD45 cluster size **d** and (3) the depletion distance **e** in the cases of PCF calculated between (1) TCR and TCR ($g_{11}$), (2) CD45 and CD45 ($g_{22}$), (3) TCR and CD45 ($g_{12}$) molecules respectively. Plots shown are for PCF calculated for areas of cells spread over anti-CD3 (green, $n = 38$), anti-CD45 (red, $n = 16$) and anti-CD11 (blue, $n = 15$)

TCR and CD45. Interestingly, coating coverslips with a lower concentration of αCD3ε (UCHT1; 1 µg ml$^{-1}$) abrogated both cell spreading and the KS (Supplementary Fig. 5E, F).

We conclude that KS occurs in both CD4$^+$ and CD8$^+$ T cells. It requires efficient spreading of the cells on coverslips, such as on αCD3 and αCD11a. It also occurs on coverslips that do not stimulate the TCR well (either coated with αCD45 or lower concentrations of αCD3ε), yet with a significantly lower extent and slower dynamics ($p \ll 0.001$).

**New forming contact regions at microvilli tips**. Our SMLM results suggested that KS occurs in early T cell contacts that are formed by the expanding tips of flexible PM protrusions called microvilli. Indeed, TCR clusters and adhesion molecules have been found at the tips of T cell microvilli[26]. To test this hypothesis, we measured the topography of the T cell PM using AFM at its apical surface and under physiological conditions. To minimize the AFM effect on the cell membrane, we used the Quantitative Imaging (QI™) mode of the NanoWizard®3 AFM (JPK), a high force mapping technique. In this technique, force curves are measured on individual pixels as follows. The AFM tip (of 2 nm radius) is first brought into contact with the PM, and is moved downwards until a certain predefined counter-force is reached. This defines the contact $z$-position. The AFM cantilever (and tip) are elevated after each force curve, such that effectively no spatial force is exerted on the cell surface (Supplementary Fig. 6A)[27]. Aside from the topography of the cell surface, this method also yielded the elasticity of the membrane, $E$ (namely, Young's modulus or the rigidity modulus), through measurement of the indentation distance of the cantilever tip into the material, $\delta$, (Supplementary Fig. 6B) and the relation[28]:

$$E + \frac{\pi(1-\nu)^2 \times F}{2 \times \tan(\varphi)} \times 1/\delta^2, \qquad (1)$$

where $\nu$ is Poisson's ratio, $F$ is the measured force and $2\varphi$ is the opening angle of the cone tip (see Methods, Microindentation analysis). The mean value of the cell rigidity, $E$, was 7.75 ± 3 pN nm. Note that we have translated Young's modulus here to rigidity modulus, following Deserno et al.[29], for consistency with modelling and simulations that we introduce below.

Because the cell size is in the range of microns (e.g. Figs. 1a and 3a), its topography image (Fig. 3a, left) could not reveal the existence of the microvilli, having a typical length of 100–200 nm (as evident by scanning electron microscopy[30]; albeit longer PM protrusions have been reported[31]). Still, using local smoothing (see Methods, Topography) we could clearly identify isolated elongations of the PM in the range of 0–200 nm (Fig. 3a, middle). To study their typical diameter we plotted their contours (Fig. 3a, right), and then, the height of the contours vs. their diameter (Fig. 3b). This representation captures the projection of the $z$ axis on an $xy$ surface. Using this representation, we defined the typical elongation head diameter of the protrusions as where the protrusion curvature inverts (i.e. where the derivative of the elongation crosses zero for the first time from the top of the protrusion). We found a typical elongation height of ~160 nm and a diameter of ~200 nm (Fig. 3b, dashed black line). To check the significance of these results, we repeated the analysis on a flat modelled membrane undergoing thermal fluctuations. We found marked differences between the protrusions from the simulated flat membrane with typical elongation height of ~10 nm and diameter of ~100 nm (Fig. 3c, d). This shows that the found protrusions do not originate from random thermal fluctuations of the membrane.

Our identification that first contacts occurring at the tips of microvilli, and their characterization using AFM at the apical

membrane of T cells, led us to visualize the 3D patterning of the segregation between the TCR and CD45 at the close contacts with the adhering part of the PM. For that, we turned to 3D PALM/dSTORM[32] of TCRζ–Dronpa and CD45 stained with Alexa647 in cells adhering on αCD3-coated coverslips that were fixed after 4 min (Fig. 3e, f; see Methods, Sample preparation). Strikingly, we observed that TCRζ and CD45 segregated not only in 2D (Fig. 3f top view, 3G), but also in their height, relative to the coverslip (Fig. 3f, side view). This segregation was ~67 nm in the close contact, although the $z$-positioning of the molecules at the PM had a relatively wide distribution of ~±22 nm (Fig. 3h). This $z$-separation across whole contacts of multiple cells was similar (~88 nm; Supplementary Fig. 6C).

We next studied the dynamics of the contacts of those membrane protrusions in spreading T cells. For that, we labelled the membrane with a fluorophore (Vybrant®DiD) and imaged the process of spreading in living cells (Fig. 3i). As with the depletion zone analysis, we selected isolated forming regions and followed them over 100 s. We found that these regions formed with an initial diameter of ~150 nm, which grew to ~2 µm over ~40 s (Fig. 3j). This dynamics is comparable to the KS dynamics we observed in early contacts (Fig. 2), suggesting that these contacts are the growing tips of touching and spreading microvilli.

**Hybrid simulations capture the TCR–CD45 segregation**. We next wanted to utilize our physical measurements of cell contacts to test our understanding of the KS process. Specifically, we aimed to recapture this process and its dynamics through predictive modelling and simulations of the cell membrane under the experimental conditions. For this, we adapted a previously published statistical–mechanical model for T-cell activation[9], and tested its ability to qualitatively capture our experimental observations, namely the physical separation of the CD45 from the TCR and its dynamics under various TCR-stimulating and non-stimulating conditions[9,33]. Briefly, in this model we embedded proteins, namely TCRs and glycoproteins (e.g. CD45, CD43 and CD148) in a physical model of the PM (Fig. 4a). The proteins could diffuse and interact on a fine grid (10 nm per pixel). A computer simulation then served to predict the cell state, based on a set of initial conditions taken from measurements. The simulation included a model that captured the energetic of the plasma membranes (PM) of interacting T cell and APC[33] (Fig. 4a, middle). Specifically, it balanced forces of repulsive interactions between glycoproteins and the PM, attractive interactions (e.g. between the TCR and anti-CD3 and the CD45 and anti-CD45), and thermal fluctuations. Initial conditions were drawn from a combination of PALM and dSTORM imaging of individual proteins (e.g. Fig. 2a) at the PM of a T cell. A molecular distribution of ligands was assumed at the engaged surface and we then let the PM of the T cell equilibrate (Supplementary Fig. 7, left grey panel). The simulation of initial cell contacts then allowed molecules to diffuse, interact and depart while the interface between the cells evolved. The TCR positions were updated from the experiment, but position of CD45 and other glycoproteins and the PM morphology (height) were simulated.

The simulation resulted in a redistributed pattern of molecules that was embedded within the interface (Fig. 4d, e). Strikingly, the simulations recreated realistic patterning of CD45 molecules around the evolving TCR clusters, that well correlated with their experimentally imaged positions on either αCD45- or αCD3-coated coverslips (compare left and right columns in Fig. 4d, e). Statistical measures were derived for the evolved patterns, focusing on the TCR–CD45 depletion distance (Fig. 5). Through these statistics, the simulation could capture both the appearance

of the KS of TCR and CD45, and its physical characteristics (Fig. 5a, c). Importantly, we could now use different physical characteristics of the simulated interface that are hard (and sometime impossible) to measure and check their effect on the KS dynamics. Our parameters included the membrane rigidity and contact radius on anti-CD3-coated coverslips (Fig. 5b), and also these parameters and the ligand density on anti-CD45-coated coverslips. We found that these parameters could dramatically influence the KS through the resultant depletion zone. To quantify the errors in the predictions of the simulations, we

calculated the RMS error between the simulated and measured depletion zones (e.g. coloured vs. black curves in Fig. 5a, c). Figure 5b, d shows these RMS errors as a function of a range of the specified physical parameters. In these figures, we highlight the set of physical parameters that were most predictive of the experimental measurements (orange contours in Fig. 5b, d). Thus, our physical model and simulations can provide a critical test of our understanding of T-cell activation by signalling complexes, and can serve to refine iteratively both experiments and models in future studies.

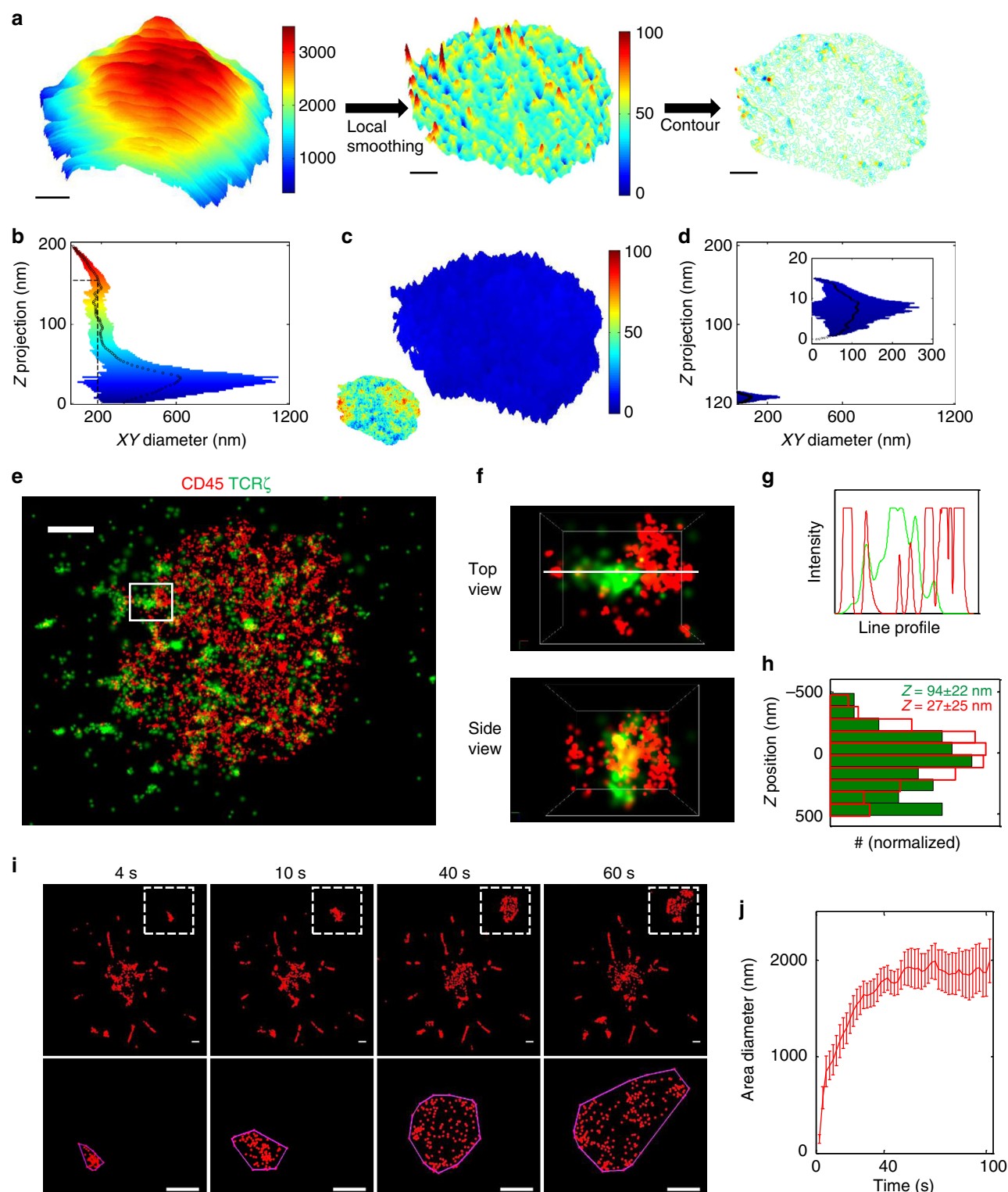

**Intricate TCR–CD45 positioning promotes localized activation.** The question arises whether KS and the creation of a depletion zone affects TCR triggering. The KS model suggests that the segregation of CD45 molecules from the TCR facilitates TCR triggering through the robust phosphorylation of tyrosines on ITAM motifs of the intracellular CD3 and ζ chains by LCK, further leading to Zap70 recruitment and downstream signalling[12]. To address this question we introduced a marker for phosphorylated TCRζ (pTCRζ) and performed three-colour SMLM. Our SMLM combined PALM imaging of TCRζ–Dronpa, and dSTORM imaging of CD45 (as described above) and pTCRζ (see Methods, localization microscopy; note differences in our imaging approach from previous work that employed either (d)STORM, e.g. refs. [34,35], or (F)PALM, e.g. refs. [36,37]). We performed the three-colour SMLM imaging of cells that were stimulated for 4 min on coverslips coated with either TCR-stimulatory (anti-CD3) or non-stimulatory (anti-CD45) antibodies (Fig. 6a, b).

We next analysed the effect of the segregation between CD45 and TCR on TCR phosphorylation (Supplementary Fig. 8A). For that, TCR molecules were given two scores: (1) Distance from CD45 molecules, indicating the extent of depletion for individual TCR molecules (Supplementary Fig. 8B) and (2) distance from pTCRζ, indicating the activation state of the same individual TCR molecules. The larger this latter distance, the molecules were considered less activated; i.e. triggering was less likely to occur in that area (Supplementary Fig. 8C). A correlation map between the two scores revealed that most TCR molecules that had the two scores in a relevant range (<500 nm), were in short distances (mostly <~50 nm) to both CD45 and pTCRζ (Supplementary Fig. 8D, E). This outcome implies that the overall state of the TCR in the synapse was more complicated than suggested by the KS model, as this model predicts a negative correlation between the scores.

Our ability to image three molecular species at once, allowed us to characterize complex multimolecular interactions between these species. Unfortunately, complex trivariate interactions cannot be analysed by standard bivariate second-order statistics (as above), since such statistics consider only two species at a time. To overcome this limitation, we applied conditional second-order statistics[37]. These statistics allowed us to quantify any increase or decrease in the propensity of two molecules to be adjacent to one another in the presence of a third molecule (Fig. 6c). For that, we distinguished two TCR populations: (1) 'Remote' TCR molecules localized 200–250 nm from a pTCRζ, which were considered as not activated (or less activated than randomly selected TCRs) and (2) 'Proximal' TCR molecules, localized under 40 nm from pTCRζ. These TCRs were considered more activated than random TCR molecules (Fig. 6d). When studying the molecular density, there were higher values of pTCRζ and proximal TCRs on the anti-CD3-coated coverslips than on anti-CD45-coated coverslips (Supplementary Fig. 9A). Note that in Supplementary Fig. 9A, the absolute molecular counts cannot be compared between the different species since pTCRζ and CD45 were captured by dSTORM. Still, interestingly, the abundance of CD45 molecules was almost three times higher for anti-CD3-coated coverslips than for anti-CD45-coated coverslips, which might be related to active recruitment of CD45 to the early contacts. The PCFs of the remote and the proximal TCR subpopulations and CD45 molecules were calculated over a wide rectangular area inside the cell footprint (Supplementary Fig. 9B), and revealed, conversely to the KS model, that the proximal TCRs were in higher correlation with CD45 at short distances than the remote TCRs (Fig. 6e). Both the SBPCF normalized by the NI model (SBPCF$_{NI}$) and by the RL model (SBPCF$_{RL}$) indicated that all TCR molecules (either remote, proximal or randomly selected TCRs) co-localize significantly with CD45 at distances below ~200 nm (Supplementary Fig. 9C, D; see Methods for the significance determination related to the PCF statistics here and below). Still, TCR co-localization with CD45 was more significant for the proximal TCRs than for all other TCRs. Overall, the SBPCF curves showed similar trends for αCD3- and αCD45-coated coverslips, but to a much lower extent on anti-CD45-coated coverslips (Fig. 6f and Supplementary Fig. 9C, D). Finally, we show that these results were not sensitive to the exact choice (i.e. distance threshold) of remote and proximal TCRs, since similar conclusions were drawn using a continuous distance score for the TCRs and BPCF analyses (see Supplementary Fig. 9E, F and Methods, Statistics).

Our three-colour SMLM imaging and analyses of the organization of TCRζ, pTCRζ and CD45 required relatively wide cell footprints, so that enough molecules from each species could be detected. Nevertheless, KS of TCR and CD45 may lead to T-cell activation within seconds from first contact. Thus, we repeated these experiments for cells fixed after 1 min from dropping onto αCD3-coated coverslips. At this earlier time-point, we could also detect the enrichment of pTCRζ in TCRζ molecules proximal (yet segregated) from CD45, as indicated by cell images (Supplementary Fig. 9G) and statistics [Supplementary Fig. 9I (right); albeit being noisy due to the low counts of detected molecules (compare Supplementary Fig. 9A, C and H, I)]. Note that our statistical analyses for the three-colour imaging (Fig. 6) are insensitive to the occasional non-uniform staining of TCRs by the αpTCRζ antibody and the endogenous population of TCRs in the cells, since neither of them depend on the placement of CD45 relative to TCR molecules.

---

**Fig. 3** New forming contact regions at microvilli tips. **a** (left) A topographic image of the cell is provided by AFM Quantitative Imaging (QI$^{TM}$), exploiting the contact point $z$ position in each pixel. (middle) A topographic image of small feature up to 200 nm is extracted after local smoothing (see Methods, Topography for details). (right) A contour analysis of the image in **a** (middle) comprising 100 contour lines with 2 nm step size. Scale bar 1 μm. **b** Mean value diameter (black circles) of the contour lines from the previous panel. Broadening represents the STD. Colour-coding as in **a** (middle; red represents high features of 200 nm and dark blue represents low features). Dashed line mark the position where the derivation of this function is zero, representing the head of the microvilli. **c** Topographic image of a simulated thermal fluctuating membrane with rigidity value of ~100 pN nm. Colour-coding identical as in **a** (middle). Inset shows the contour analysis. **d** Mean value diameter (black circles) of the contour lines analysed for the thermal fluctuating membrane. Broadening represents the STD. Colour coding is identical to the one used in **a** (middle). Inset shows enlargement of the area selected. **e** Two-colour SMLM, PALM combined dSTORM of Jurkat cells expressing TCRζ–Dronpa (green) and CD45 immunostained with Alexa647 (red) spread on an anti-CD3-coated coverslip 4 min before fixation. Scale bar 2 μm. **f** Zoom on a region containing a TCR cluster in **e**. Top view (upper panel) and side view (bottom panel) are presented. **g** The intensity profile across the white line in **f**. **h** Histogram of $z$ heights of detected molecules in **f**. TCR represented by full green bars and CD45 by empty red bars. **i** Upper panel, time sequence (at 4, 10, 40 and 60 s) of the synapse formed in between Did fluorophore membrane tagged Jurkat cell and anti-CD3-coated coverslip. Lower panel, zooming into the distinct area selected with convex envelope surrounding the membrane molecules. Scale bar 0.5 μm. **j** Convex envelops diameter of distinct areas ($n = 24$) as a function of spreading time. $t_0$ is set as the appearance time of molecules in the areas

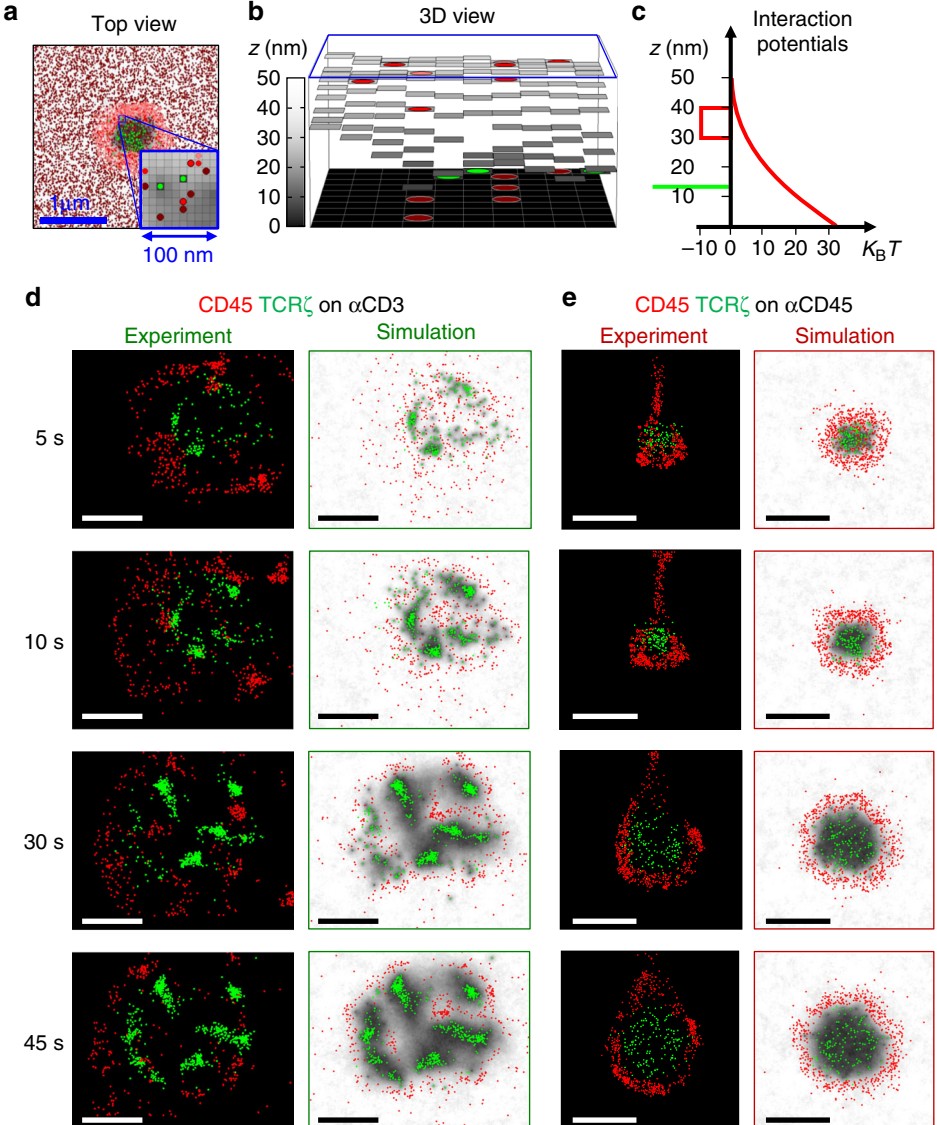

**Fig. 4** Numerical modelling and hybrid simulations capture the KS at the tips of microvilli. A numerical model that captures the energetics of the plasma membranes (PM) of interacting T cell with an activating coverslip. **a** Top view of a representative simulated interface at $t = 10$ s. Dark red points show anti-CD45 ligands randomly distributed with density = 1000 molecules $\mu m^{-2}$. Red points show locations of CD45, light red points show locations of other glycoproteins (GP), namely CD43 and CD148. Green points show locations of TCRs. The inset shows an enlarged view of a 100 nm × 100 nm square. The three types of proteins and the dark red ligands are shown. The grey levels mark the height of the membrane from 0 to 50 nm. **b** 3D in scale view of the inset in **a**. Patterning of the height of the PM is illustrated in the $z$ axis due to thermal fluctuations and molecular interactions. **c** The potential energy ($V$) of molecular interactions at the PMs of T cell and APC (or activating coverslip), as a function of their local separation ($z_{ij}$). The thin green rectangle is effectively infinite in length as the membrane at the TCRs is forced to be $z = 13$ nm. The red rectangle between 30 and 40 nm is the interaction potential of the CD45 and anti-CD45. The red line describes the repulsive energy of the CD45 and the other glycoprotein (GP; CD43 and CD148) molecules. **d** (left) Experimental SMLM results, as shown in Fig. 2a, of newly forming contacts of E6.1 cells spreading on anti-CD3ε-coated coverslips. Green points are TCRs and red points are CD45. Frames are taken at 5, 10, 30 and 45 s. (right) Simulated results. In the anti-CD3 simulations we did not use 'ligands' but instead forced the membrane at the locations of the TCRs to a height of 13 nm. **e** (left) Experimental SMLM results, as shown in Fig. 2a, of newly forming contacts of E6.1 cells spreading on anti-CD45-coated coverslips. (right) Simulated results. The simulation ran with ligand density = 1000 molecules $\mu m^{-2}$, rigidity = 25 $K_B T$ and TCR–CD45 maximal apparent separation = 200 nm. The scale bar is 1 μm

Finally, we developed a new analysis to study the morphology of TCR clusters and their organization in relation to CD45 clusters in the synapse. First, a contour of the TCR clusters was drawn, which was then used to draw a topological skeleton map of the clusters under study (Fig. 7a). This map emphasizes morphological properties of the cluster shape, and represents the ridgeline of the density map of TCR molecules. Next, we calculated univariate PCFs of the previously identified remote and proximal TCRs. However, instead of applying growing rings as a basis for the spatial statistics, we used dilation shapes of the original skeleton that were applied iteratively on the same map (Fig. 7b). Naturally, since all molecular species under study spatially relate to TCR clusters, the correlation was positive and decreased with the extent of dilation. Nevertheless, the two distinct TCR populations behaved differently, as the proximal TCRs correlated better with the ridgeline than the remote TCRs (Fig. 7c). As expected, the correlation normalized by the RL model did not show any segregation between the populations and

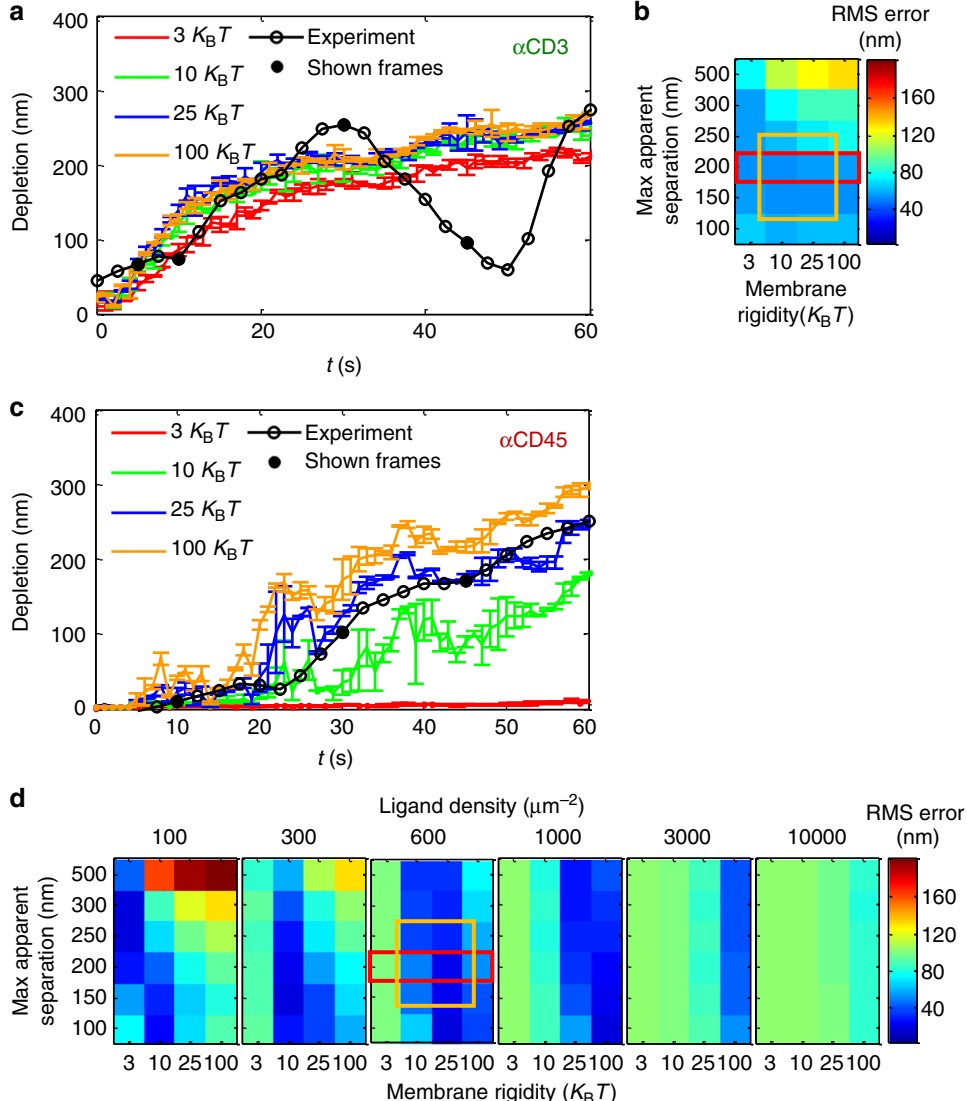

**Fig. 5** Modelling and simulations serve to quantify the effect of physical properties on the KS extent and dynamics. **a** The depletion distance of simulated cells on αCD3ε-coated coverslip, as a function of time for a range of simulated rigidities. **b** The RMS error (in nm) of the simulated and experimental depletions under a range of rigidity and contact radii. Highlighted in red are the shown conditions in **a**. Highlighted in orange are the conditions that showed the most robust results (for stimulated and unstimulated cells). **c** The depletion distance of simulated cells on αCD45-coated coverslip, as a function of time for a range of simulated rigidities. **d** The RMS error of the simulated and experimental depletions under a range of rigidity and contact radii (within single matrices) for a range of αCD45 ligand densities (different matrices). Highlighted in red are the shown conditions in **c**. Highlighted in orange are the conditions that showed the most robust results (for stimulated and unstimulated cells)

the ridgeline (Fig. 7d). Notably, similar results were obtained using analyses that delineated CD45 clusters (Supplementary Fig. 10). This further indicates the robustness of our morphological analyses and related conclusions.

## Discussion

The KS model has been proposed to facilitate T-cell activation by the TCR, but has not been resolved and studied at the nanoscale. This has limited our understanding of physical characteristics related to the KS model, its early dynamics, its functional role in relation to TCR triggering within nano- and micro-scale clusters, and its relation to alternative mechanisms that have been suggested for antigen discrimination by the TCR[15]. Here, we developed an approach for SMLM imaging of the TCR and CD45 at the PM of live cells. Using this technique, we detected the KS of these two proteins with an unprecedented spatial resolution of

20–30 nm and in single molecule detail. We could further follow the dynamics of this process with and without cell activation, with temporal resolution down to 2–3 s per frame. Importantly, our PALM/dSTORM imaging aimed to resolve the KS between the TCR and CD45 and was not optimized for the absolute counting of these molecules. Such counting is imprecise, esp. for live cell imaging and for dSTORM[38].

As expected from the KS model, the dynamic physical separation between the TCR and CD45 occurred during synapse formation on TCR stimulating coverslips. We further observed that KS mainly formed at remote small regions that spread till fusing together with the main part of the synapse. We showed that the segregation was rapid and robust. It started from a minimal size of 50–100 nm at the earliest observation time and reached a maximal size of ~230 nm over a few dozens of seconds, before the molecules got mixed together eventually. Surprisingly, yet consistent with Chang et al.[13], the KS between the TCR and

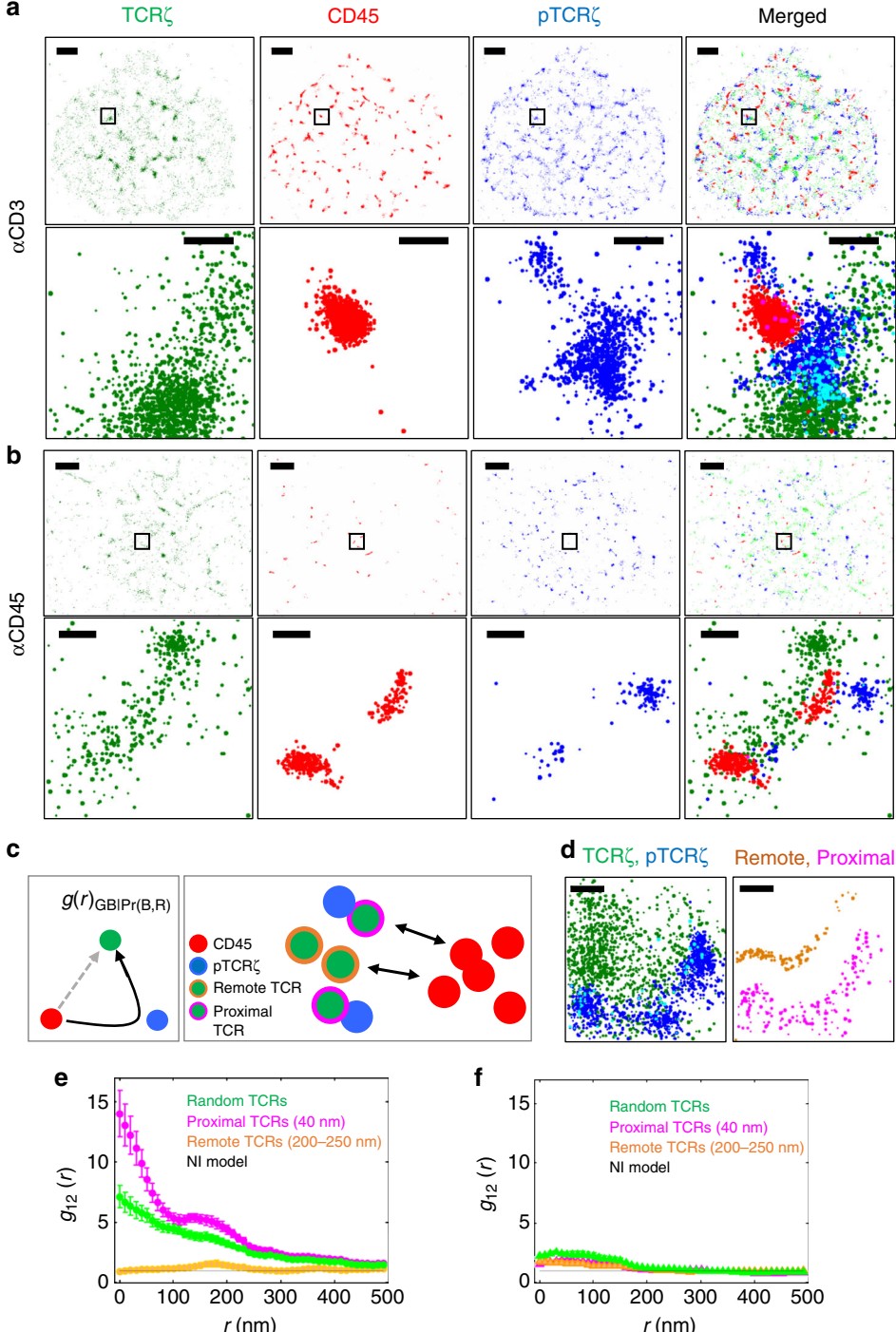

**Fig. 6** TCR phosphorylation is enriched at TCR cluster edges that are proximal, yet separated from CD45. **a**, **b** Multicolour SMLM, PALM combined dSTORM images of Jurkat cells expressing CD3ζ-Dronpa (green) and immunostained for CD45 (red) and pTCRζ (blue) with Alexa568 and Alexa647 correspondingly, fixed after 4 min of spreading on an anti-CD3 (**a**) or anti-CD45 (**b**) coverslips. Scale bar 2 μm. Lower panels show enlargement of the marked areas, scale bar 200 nm. **c** A conditional bivariate PCF analysis between TCR and CD45 molecules as a function of TCR-pTCR distance (left). (right) Illustration of the process, TCR (green) are categorized into subpopulations of TCR: 1. Proximal TCR (green with magenta outlined) with a distance < 40 nm to pTCR (blue). 2. Far TCR (green with orange outlined) with a distance of 250 < d < 200 nm to pTCR (blue). Then PCF is calculated to the subpopulations with CD45 (red). **d** An example of the process, as a function of the distance from pTCR (blue), Proximal TCR (green) are coloured with magenta and far are coloured with orange, scale bar 200 nm. **e**, **f** Conditional bivariate PCF analyses between TCR and CD45 molecules as a function of TCR-pTCR distance of Jurkat cells fixed after 4 min from dropping on **e** anti-CD3-, or **f** anti-CD45-coated coverslips

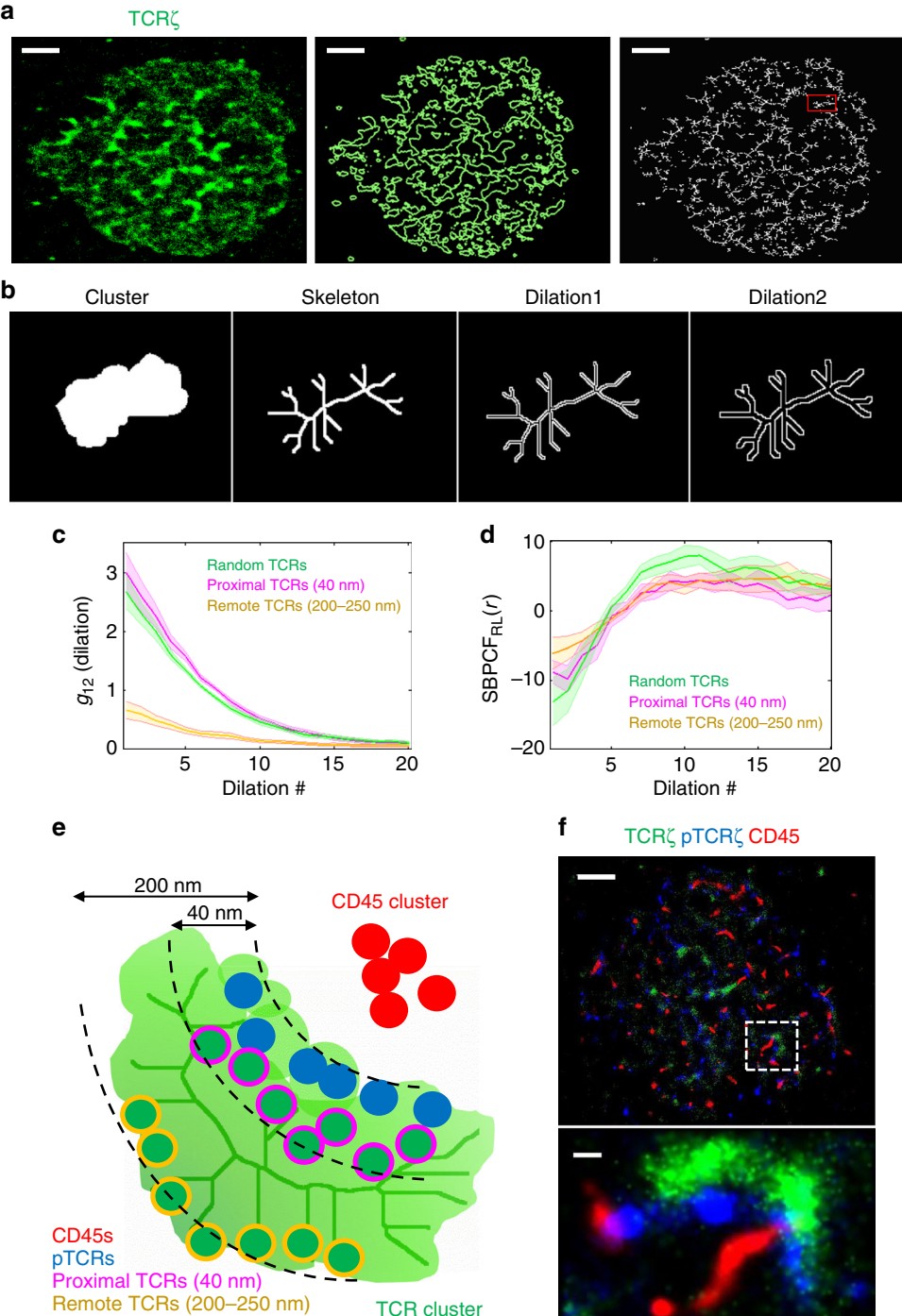

**Fig. 7** Intricate positioning of TCRs and CD45 promotes localized TCR activation within clusters. **a** (left) Scatter plot of TCR molecules of a representative Jurkat cell fixed after 4 min from dropping on an anti-CD3-coated coverslip. (middle) Contour map (contour level 2% of all contours) of TCR density map. (right) Skeleton map of the contour generated by Matlab morphological operations tool box, BWMorph. Scale bar 2 μm. **b** The process of dilation. An isolated area (section in red in **a** right) is iterated with dilations. **c** A conditional bivariate PCF analysis adopting the dilation shapes as a basis for the spatial statistics between TCR and CD45 molecules of Jurkat cells fixed after 4 min from dropping on anti-CD3-coated coverslip. **d** The conditional bivariate PCF shown in **c** normalized by the RL model. **e** Illustration of the model proposed. **f** A representative Jurkat cell fixed after 4 min from dropping on an anti-CD3-coated coverslip. Scale bar 2 μm and 200 nm in the enlarged view

CD45 occurred also on non-stimulating coverslips, coated by either αCD45 or αCD11a. We conclude that KS requires efficient spreading of the cells on coverslips, such as on αCD3 (UCHT1) and αCD11a. It also occurs on coverslips that do not stimulate the TCR well [either coated with αCD45 or lower concentrations of

αCD3 (UCHT1], yet with a significantly lower extent and slower dynamics.

The observation of the KS in separated contacts led us to hypothesize that they form at the tips of engaged microvilli. Indeed, using dSTORM microscopy, Jung et al. have found TCR

clusters at the tips of T-cell microvilli[26]. Here, we turned to AFM to study the topography of the PM at the apical side of live cells, as they adhered on coverslips. Using this technique, we found the typical dimensions of microvilli and the rigidity of the PM. Three-dimensional PALM/dSTORM imaging allowed us to resolve the segregation of TCR and CD45 in close contacts of the cells with TCR-stimulating coverslips. This imaging revealed that these molecules segregated both in the plane of the coverslip and perpendicular to it, indicating coincidence of the KS with PM height differences, as expected at microvilli tips and PM ruffles. We further studied the dynamic enlargement of the contacts upon cell spreading, by SMLM imaging of membrane dyes (DiD). Strikingly, the tip dimensions and their dynamic enlargement matched the separated contact sites that showed KS, thus implicating them as the early contacts that mediate early T-cell activation.

From our results, we conclude that KS is driven by a combination of mechanical forces of molecules and the PM, as predicted by the KS model. To quantitatively evaluate these forces, we adapted a computational model and simulations for early T-cell activation[5]. We then performed Monte-Carlo hybrid simulations in which positions of TCR molecules were taken from SMLM movies, while the patterning of the PM and CD45 (and other glycoprotein) molecules were predicted from the simulations. Strikingly, our modelling and simulations could capture the KS and the resultant depletion distance and its dynamics under either TCR activating or non-activating conditions. Thus, our simulations provide an elaborate test of our physical understanding of the KS process and its underlying forces. These simulations could further serve to quantify physical parameters that are not easily accessible by experiments, such as the PM rigidity, ligand density and the size of the microvilli edges. Importantly, the simulations could capture the effects of realistic changes in these parameters on the KS size and dynamics. Thus, the simulation could become an invaluable predictive and hypothesis-generating tool in the study of activation of T cells and other cells of interest.

The KS model suggests that the segregation of TCR from CD45 is crucial for TCR triggering. Here, we showed that phosphorylated (and thus, triggered) TCRs were conversely, and surprisingly, more correlated to CD45 molecules in comparison to other TCRs. Moreover, pTCRs localized in close proximity to the ridgeline of TCR clusters. This explains the localized patterns of TCR activation that we and others previously observed[9,10]. Importantly, this also relieves a persistent puzzle and criticism regarding the KS model[39]—on one hand, CD45 is needed for TCR activation via Lck dephosphorylation and activation, yet it should also be removed from TCRs to block its maintained quenching of TCR activation via its phosphatase activity. Our results show that within the initial adhesion contacts, at the tips of engaged microvilli, the CD45 cluster is somewhat separated from the TCR cluster, yet remains in very close contact (Fig. 7e, f). At that point, dephosphorylation of both Lck and TCR ITAMs can continuously occur, thus leaving Lck in its active form, but quenching the TCR signal. If the TCR is engaged under stimulating conditions, a tight bond is formed between the TCRs, already in clusters, and their ligands. Consequently, the two opposing membranes of the T cell and of the APC are brought into close apposition. Then, within a few dozens of seconds and because of mechanical tension over the T-cell membrane, CD45 molecules are segregated from the close contact between the membranes, leaving a depleted zone of ~250 nm. This separation is sufficient for active Lck to fully activate the pMHC-TCR complex, and the signalling cascade is bound to proceed. Indeed the separation of Lck from CD45 and its colocalization with

pTCRs in activated cells, but not in resting cells, have been previously captured by SMLM[40].

Taken together, our results suggest a refined view of the KS model where it occurs within seconds from TCR activation at the engaged tips of microvilli, and that TCRs should be segregated, yet not removed too far, from CD45 for their optimal activation within clusters. Detailed computer modelling and simulations of the IS has allowed us to recover the fast dynamics and physical properties of the observed KS, providing a critical test of our physical understanding and a quantitatively predictive tool for future studies. Thus, our results shed new light on molecular mechanisms that govern T-cell activation and set the stage for studying the KS role in health and disease. Our imaging and computational techniques are relevant to studying additional membrane proteins in various cell types.

## Methods

**Cloning.** TCRζ chain was tagged with the PAFP Dronpa[19] (MBL International Corporation) was cloned in EGFP-N1 or EGFP-C1 vectors (Clontech). PAFP gene was cut from a vector through digestion with restriction enzymes (typically AgeI and NotI, XbaI or BsrGI). PAFP gene then served to replace existing fluorescent proteins (FP) in previously used constructs using similar digestion reactions and ligation of the PAFP insert (Quick-ligation kit, New England BioLabs). Validation of cloning was done by restriction digestion analyses and DNA sequencing of the inserts.

**Cell lines.** Jurkat E6.1 (CD4$^+$) T cells were a kind gift from the Samelson lab at the NIH and Jurkat J76 (CD8$^+$) cells were a kind gift from the Acuto lab at Oxford. These cells were transfected with DNA using a NEON electroporator (Invitrogen) for the expression of proteins tagged with PAFPs. Lines of Jurkat E6.1 cells, stably expressing TCRζ–Dronpa, were available for this study from previous work[7]. Briefly, in that work the cells were created by selection with Geneticin at 1.5 mg ml$^{-1}$ (G418, Invitrogen). After 2–3 weeks, the cells were sorted and single clones were grown in 96 well plates. After 3 additional weeks, the extent of protein expression was checked by flow cytometry. Cells were then evaluated using biochemistry assays, flow cytometry, confocal microscopy (510 LSCM, Zeiss) and epifluorescence, TIRF and PALM imaging, as described below.

**Sample preparation.** The preparation of coverslips for imaging spread cells followed a previously described technique[18]. Briefly, for diffraction limited and PALM imaging, coverslips (#1.5 glass chambers, LabTek or iBidi) were washed with acidic ethanol at room temperature (RT) for 10 min; liquid was then aspirated and coverslips were dried at 37 °C for 1 h. Cleaned coverslips were incubated at RT for 15 min with 0.01% poly-L-lysine (Sigma) diluted in water. Liquid was aspirated and coverslips were dried at 37 °C for 12 h. Coverslips were subsequently incubated with stimulatory or non-stimulatory antibodies at a concentration of 10 μg ml$^{-1}$ (unless specified otherwise) overnight at 4 °C or 2 h at 37 °C. Finally, coverslips were washed with PBS. Throughout the study we used the following stimulatory antibodies: purified mouse anti-human CD3ε (clone UCHT1, Affymetrix eBioscience, 160038) and purified mouse anti human CD45 (Bactlab Diagnostics, PMG555480) and mouse anti-human CD11a (BD Pharmingen, 555378) as non-stimulatory antibodies. A few hours before imaging, cells were resuspended in imaging buffer at a concentration of 1 million/150 ml and 100,000–500,000 cells were dropped onto coverslips for PALM or dSTORM imaging, incubated at 37 °C for the specific spreading time (typically 3 min) and fixed with 2.4% PFA for 30 min at 37 °C or used for live cell imaging.

For colour aberration and imaging accuracy characterization, 100 nm Tetraspec beads (Invitrogen Inc.) were diluted 1/1000 (v/v) in TDW and deposited on a bare coverslip.

**Immunostaining.** CD45 proteins were labelled using dye-labelled goat anti-mouse secondary antibodies. Antibodies were used following the manufacturers' protocols. In brief, 0.5 μg of mouse anti human (anti-CD45 monoclonal antibody) was added to 500 × 10$^3$ cells suspended in FACS buffer for 30 min on ice. Then cells were washed two times in phosphate buffered saline (PBS) and suspended in 1.5 ml of FACS buffer (90% PBS 10% FBS 0.02% NaAzide) containing 1 μg of goat anti-mouse IgG1 (γ1) secondary antibody, Alexa Fluor 647 conjugate (Life Technologies, A21240). Cells were washed and suspended in imaging buffer (RPMI without phenol red, 10% FBS, 25 mM HEPES).

PhsophoTCRζ proteins were labelled in fixed cells. In brief, for permeabilization, 0.4 ml of 0.1% Triton X-100 in PBS was added per well and incubated for 3–4 min. The cells were blocked by 2% normal goat serum in PFN (PBS + 10% serum + 0.02% sodium azide) for 30 min. For 0.5 million cells, we added 0.5 μg rabbit anti-human phosphoTCRζ (pY83 of TCRζ; Thermo Scientific, 700177) as a primary antibody diluted in 2% normal goat serum in PFN, incubated for 60 min at RT and washed three times with PFN. Alexa647 was added as a secondary antibody (anti-

rabbit, A21244, Life Technologies) diluted (1/3000) in 2% normal goat serum in PFN, incubated for 45 min at RT and washed three times with PFN.

Additional antibodies used for immunostaining included AlexaFluor647 (EP286Y) Anti-CD3 zeta antibody (Abcam, AB-ab197037), anti-human CD45 AlexaFluor647 (BioLegend, BLG-304056) and anti-Human alpha beta TCR purified (Biotest, 14-9986-82).

The PM was tagged by incubation of the cells in staining solution containing 10 μM DiD (Vybrant® DiD Cell-Labeling Solution, Invitrogen, V22887), in PBS for 0.5–5 min. After staining, cells were washed and suspended in imaging buffer.

**Single-molecule localization microscopy**. Two-colour SMLM (combined PALM/dSTORM) imaging was conducted both for fixed and live cells. In general, cells were suspended in a STORM imaging buffer which was made by a protocol previously described[25], and imaging was performed using a total internal reflection (TIRF) Nikon microscope. Imaging in TIRF mode served to visualize molecules at the PM of spreading cells in close proximity to the coverslip (up to ~100–200 nm). Photoactivatable fluorescent protein and fluorophores were activated using a low intensity laser illumination at 405 nm (~0.5%), and sequentially imaged in a following frame using laser excitation at 488, 561 or 647 nm. The focus of the microscope was maintained throughout the imaging using the PerfectFocus system of the microscope. PALM acquisition sequence typically took ~5 min at 50–100 fps. For two-colour SMLM imaging, we used chimeric constructs of molecules with Dronpa[7], and immunostaining of proteins with antibodies labelled with Alexa647. For three-colour SMLM we added immunostaining of proteins with antibodies labelled with Alexa568. Combined SMLM (PALM-dSTORM) imaging was in a dSTORM buffer (50 mM TRIS pH = 8, 10 mM NaCl, 0.5 mg ml⁻¹ glucose oxidase, 40 μg ml⁻¹ catalase, 10% glucose, 10 mM MEA). The fluorescent constructs were imaged with fast alteration of the imaging channel in consecutive frames. Photo-activation illumination at 405 nm was changed over the imaging sequence of fixed cells. Three-dimensional PALM was conducted using the astigmatism method[32].

**Calcium imaging**. For calcium-flux experiments, Jurkat E.61 T cells were loaded with Fluo-4AM (Molecular Probes) at 5 μM for 30 min in the presence of 2.5 mM probenecid. T cells were transferred to imaging medium (RPMI without phenol red, 10% FBS, 25 mM HEPES) and allowed to adhere to the coated coverslips. We quantified Fluo-4 responses by determining the average intensity of a region within each cell as a function of time using the ImageJ program (NIH).

**Statistical analyses**. All statistical analyses were conducted using a t-test (assuming two-tailed distribution and two sample unequal variance). Significance related to univariate and bivariate PCF data was determined by comparing the measured PCFs to null hypotheses, as mentioned in the text. The 95% confidence interval for the null hypotheses was determined by simulating 19 realizations using Monte-Carlo simulations, and by taking their maximal and minimal (univariate- or bivariate-) PCF values at each scale.

**Data processing**. Data acquired by single-molecule localization microscopy was analysed by the N-STORM module in NIS-Elements (Nikon) or a previously described algorithm (ThunderSTORM)[41] for the identification of individual peaks and group them into functions that reflect the positions of single molecules[17]. Next, peaks were grouped and assigned to individual molecules for rendering. Peak grouping used a distance threshold and a temporal gap to account for possible molecular blinking[17]. For fixed cells experiments, a temporal gap of ~50 ms and a distance threshold of 20 nm were applied for each fluorophore separately in order to minimize possible over-counting of molecules. Drift compensation and channel registration were performed using dedicated algorithms in the ThunderSTORM software. For live cells experiments no drift compensation was applied and a distance threshold of 20 nm was taken (regarding time gaps, each image in a live experiment accumulates 2–2.5 s of acquisition time as will be describe next) Three dimensional PALM was conducted and analysed using the astigmatism method[32]. Calibration was conduct using 100 nm Tetraspec fluorescent beads (Invitrogen). Three-dimensional decoding was performed using Nikon NSTORM software. Individual molecules are presented in PALM and dSTORM images with intensities that correspond to the probability density values of their fitted Gaussian with respect to the maximal probability density values detected in the field.

**Analysis and presentation of live cell PALM imaging data**. To generate a frame in a live cell movie, accumulation of 200 frames of an SMLM acquisition movie with a frame rate ranging 80–100 fps was done, with alternating acquisition of the green and red channels. Thus, each image represents 2–2.5 s of acquisition time in the SMLM acquisition. The images were assigned the frame time of the first participating frame from the SMLM movie (e.g. the presented frame for 2 s consists of frames accumulated between 2 and 4 s). These accumulated frames were further used to generate movies of the cell spreading (Supplementary Movies 1–4). The accumulated frames were next analysed in one of three ways, as described in the Results section. The images in Fig. 2 (and frames in Supplementary Movies M1, M3) were generated by filtration of the Dronpa tagged proteins, which showed relatively fast bleaching, and was done using custom built Matlab algorithm (The MathWorks, Natick, MA) with variance estimate of acquisition noise of 0.05 and

bias of the prediction of 0.5. SMLM movies (and individual frames) show individual molecules as dots. We preferred this representation over alternative representations (such as individual or summed Gaussians[17]) since it better shows sparse and newly recruited molecules, and their position relative to the cell contact. Drift correction was applied for fixed-cell imaging, but not for live-cell imaging due to the fast rate and overall short duration of our imaging (2 s per an effective PALM/dSTORM frame).

**Second order statistics and pair correlation function**. SMLM imaging results in a point pattern that marks the centre locations of single molecules. Second-order statistics is useful for the analysis and interpretation of such point patterns[22]. Here we employ the statistics of pair correlation functions (PCF, denoted also as $g(r)$), for studying the intermolecular density variations within the resultant point patterns. Univariate PCFs serve to explore point patterns of a single species, while bivariate PCFs, as defined below (Eqs. (2) and (4)) are required for studying two different species.

We construct our statistics following the Wiegand–Moloney's O-ring statistic[22]. For two point patterns that represent two different populations (denoted 1 and 2), the bivariate PCF $g_{12}(r)$ is defined as follows:

$$g_{12}(r) = \lambda_2^{-1} \begin{bmatrix} \text{the number of points of pattern 2 at distance } r \\ \text{from an arbitrary point of pattern 1} \end{bmatrix}, \quad (2)$$

where $\lambda_2$ is the mean areal density of points of pattern 2. The Wiegand–Moloney's $O_{12}(r)$ is defined as[22]:

$$O_{12}(r) = \lambda_2 g_{12}(r). \quad (3)$$

A bivariate PCF can be calculated for a pixelated image using the following definitions:

$$g_{12}(r) = \frac{A}{n_2} \frac{\frac{1}{n_1}\sum_{k=1}^{n_1} \text{Pnts}\left[S_2, R_{1,k}^w(r)\right]}{\frac{1}{n_1}\sum_{k=1}^{n_1} \text{Area}\left[R_{1,k}^w(r)\right]}, \quad (4)$$

where the operator Pnts $[S_j, X]$ counts the points of $S_j$ in region $X$. The operator Area counts the number of pixels in the region $X$. $R_{1,k}^w(r)$ is a ring with radius $r$ and width $w$ centred on the $k'$th point of type 2. $n_i$ is the total number of points of type $i$ in the study region of area $A$.

In general, we calculated the PCFs for the experimental data and compared these PCFs to different models using Monte-Carlo simulations. For that, we used a custom code in Matlab. Two useful models for the evaluation of the organization and extent of interactions between species are the No Interaction (NI) model and the Random Labeling (RL) model, which we describe below.

**The NI model**. The NI model assumes no interaction (either attraction or repulsion) between the points of the pattern. According to the NI model, the $g_{12}(r)$ function is equal to one. $g_{12}(r) > 1$ indicates aggregation (or attractive forces), while $g_{12}(r) < 1$ indicates repulsion. For a finite set of points, the $g_{12}(r)$ curves due to the NI model can deviate from a value of 1. Thus, it is useful to simulate multiple realizations (here, 19) of non-interacting species. The top and bottom $g_{12}(r)$ values at each scale of all simulated patterns can then serve to define the (95%) confidence interval due to the NI model.

**The RL model**. In order to investigate whether or not two species in a joint point pattern are significantly interacting, one could use the RL model. In this model, points of pattern 1 ($n_1$) and points of pattern 2 ($n_2$) distribute randomly in the detected ($n_1 + n_2$) locations. Multiple Monte-Carlo simulations replicate the point patterns while randomly re-labelling the points (with the number of points from each species). The bivariate PCF of the original point pattern $g_{12}(r)$ is then compare to the bivariate PCFs of the simulations. We used the lowest and highest $g_{12}(r)$ of the 19 different simulations as a 95% confidence interval for the acceptance or rejection of the model as a null hypothesis. Agreement of the data with the RL model indicates homogeneous mixing, and hence strong interaction (in a statistical sense) of the two species under study.

**Standardized bivariate pair-correlation functions**. For comparison of the bivariate functions between different time points and different stimulating conditions, these bivariate functions were compared and standardized according to the models described above by the following way:

$$\text{SBPCF}_{\text{NI}}(r) = \frac{g_{12}(r) - \text{NI}_{\text{mean}}(r)}{\text{NI}_{\text{var}}(r)}, \quad (5)$$

$$\text{SBPCF}_{\text{RL}}(r) = \frac{g_{12}(r) - \text{RL}_{\text{mean}}(r)}{\text{RL}_{\text{var}}(r)}, \quad (6)$$

where $NI_{mean}(r)$, $NI_{var}(r)$, $RL_{mean}(r)$ and $RL_{var}(r)$ are the mean and variance of the simulated patterns due to the NI and RL models, respectively.

**Depletion zone.** The BPCF function $g_{12}(r)$ of two sets of molecules can capture the appearance and dynamics of a 'depletion zone', a zone with no molecules. The depletion range is evident through $g_{12}(r) < 1$, namely the $r$ values for which the BPCF curve is smaller than the ones expected from the NI model (e.g. Fig. 2b). The BPCF may exceed the values due to the NI model at other (longer) ranges, where the effect of the depletion is no longer resolved and where the proximity of the interacting molecules mask their segregation at smaller length-scales.

**Conditional bivariate PCF.** In a recent publication, we have shown how three-colour SMLM imaging could help to quantify synergy in the interactions of three interacting species[37]. We briefly describe this analysis below, for a comprehensive discussion and robustness analyses of this statistics can be found in ref. [37]. The conditional bivariate statistics and synergy analyses rely on a first step of choosing a subpopulation of one species, based on its proximity to (or interaction with) a second species. Then, the interaction of this subpopulation with a third species can be compared to a similar interaction, yet for random sets of the first species. This analysis is applied to study regions that cover most of the apparent footprint of the cells. As a next step, the standardization of these conditional bivariate PCF curves allows for their comparison and for their averaging over multiple cells. After this step, significant synergy in molecular interactions can be detected.

We briefly describe below the algorithm by referring to an example where we study the interactions between type 1 and type 2 molecules, upon binding of type 2 to type 3 molecules. We denote the set of $x,y$ coordinates of each molecular species by $S_i$, where $i = 1,2,3$. First, we define a Boolean function $Pr(s_2,s_3)$, which identifies the proximity of molecules from two species of interest (e.g., Type 2 and Type 3). Pr is calculated for each pair of molecules $(s_2, s_3)$ from $S_2$ and $S_3$, as follows:

$$Pr(s_2, s_3) = \begin{cases} 1 & Dist(s_2, s_3) \leq d_{th} \\ 0 & Dist(s_2, s_3) > d_{th} \end{cases} \quad (7)$$

where $d_{th}$ is the threshold for defining proximity, and the operator $Dist(s_i, s_j)$ is the Euclidean distance between the points $s_i$ and $s_j$. A threshold of 40 nm was then chosen to select the interacting Type 2 molecules, using the function Pr to obtain a subset $S_2'$ of the points of type 2 (namely $S_2$), following the set-builder notation of Eq. (8):

$$S_2' = \{s_2 | \exists s_3 \in S_3 \text{ such that } Pr(s_2, s_3) = 1\} \quad (8)$$

Notably, in our SMLM measurements the localization uncertainty of individual molecules peaked at ~25 nm for all colours. Thus, the 40 nm threshold was about the rms size of the uncertainty of two colocalized molecules in our study. Another consideration for setting the proximity threshold involves the molecular density in the data (as detailed in ref. [37]).

Together, Eqs. (7) and (8) state that $s_2$ is included in $S_2'$ if there exists at least one proximal molecule $s_3$ from $S_3$ that lies below the threshold distance $d_{th}$ from $s_2$.

Next, we calculated the conditional bi-variate pair-correlation function (BPCF; $g_{12|Pr(2,3)}(r)$) of the selected subset of Type 2 molecules, $S_2'$, with a third molecular species of Type 1. This function is defined similar to the BPCF (Eq. (4)), as follows:

$$g_{12|Pr(2,3)}(r) = \frac{A}{n_{2'}} \frac{\frac{1}{n_1} \sum_{k=1}^{n_1} Pnts\left[S_2', R_{1,k}^w(r)\right]}{\frac{1}{n_1} \sum_{k=1}^{n_1} Area\left[R_{1,k}^w(r)\right]}, \quad (9)$$

Note that this equation refers to $S_2'$, the proximity-selected sub-population of $S_2$, with a total number of molecules of $n_{2'}$.

**AFM data acquisition and analyses.** Measurements were carried out using commercial AFM a Nano-Wizard 3® AFM (JPK Instruments AG) equipped with temperature controller coverslip holder ('Biocell', JPK Instruments AG) mounted on an Eclipse Ti-E microscope (Nikon Instruments). Silicon Nitride (Si3N4) AFM cantilevers with sharp silicon tips (MSNL10, nominal tips radius ~2 nm, Bruker) were cleaned and oxidized using $O_2$ Plasma (Atto, Diener Electronic) for 5 min prior to use. Cantilever force constants ranged from 0.009 to 0.05 N m$^{-1}$ as determined by individual calibration of the cantilever using the thermal noise method. To obtain the distribution of the properties over the cell surface and simultaneously record cell topography with minimal shear forces, the quantitative mode (QI mode®, JPK Instruments AG) of operation was utilized. Images of cells were collected with the resolution of $512 \times 512$ pixels (typically within $20 \times 20$ μm$^2$ area). Force curves were set with the following parameters: setpoint force of 130 pN, Z length of 60 nm, speed of 50 μm s$^{-1}$ and with a frequency of 100 kHz. Sample preparation was identical to what is described above with the following changes: for better adhering the cells to the coverslip, the coverslip was modified with all three stimulatory and non-stimulatory antibodies simultaneously: purified mouse anti human anti-CD3ε (clone UCHT1), purified mouse anti human anti-CD45 and purified mouse anti human anti-CD11a.

**Microindentation analysis.** Microindentation is a method when an indenter with well-defined geometry punch into the cell. The indenting force and the resulting indentation in cells often follow the prediction of the Hertz model[28]. Young's moduli or rigidity moduli of cells can be calculated from the force–indentation curves by fitting them to the Hertz model. Contact point coordinates $Z_0$ and $D_0$ (Supplementary Fig. 6A,B) was calculated utilizing the commercially available data processing software of JPK. Then the sample deformation $\delta$ and indenting force $F$ are calculated as:

$$\delta = \begin{cases} 0 & z < z_0 \\ (z - z_0) - (d - d_0) & z \geq z_0 \end{cases} \quad (10)$$

$$F = \begin{cases} 0 & z < z_0 \\ k(d - d_0) & z \geq z_0 \end{cases} \quad (11)$$

Then linear fitting is applied to fit the $F$ vs. $\delta^2$ data in the post-contact region, $z \geq z_0$ due to the Hertz model to extract Young's modulus, $E$ of the cell:

$$F = \frac{2E \times \tan(\varphi)}{\pi(1 - V)^2} \delta^2, \quad (12)$$

where $v$ is Poisson's ratio and $2\varphi$ is the opening angle of the cone tip. Finally the transition to rigidity moduli was done following Deserno[29]:

$$\kappa = \frac{1}{48} \times E \times h^3, \quad (13)$$

where $h$ is the membrane thickness and was taken to be 4 nm.

**Topography image analysis.** For revealing the fine details, i.e. 100 nm elongations over the micron scale topographic image, Img, using Matlab as the processing tool, the following steps were carried out:

Because the AFM measurement is built by measuring line by line it is necessary to average each line by a moving window separately, in particular when the measurement acquired over a long period of time.

1. The lined smooth Img (line span = 5 pixels, column span = 0 pixels) was subtracted from the original:

$$ImgLineS = Img - smooth(Img, 5, 0) \quad (14)$$

2. A simple average by a moving window was done (moving window span = 8 pixels) using the smoothn Matlab addon function:

$$ImgS = smoothn(ImgLineS, 8) \quad (15)$$

3. A contour map was provided by dividing the ImgS into 100 height lines.
4. The projection of the height ($z$ axes) vs. $xy$ diameter plot was provided by plotting each contour line height by its diameter. Averaging is due to the fact that there are several contour lines for a given height value.
5. We assumed by previous observations, the microvilli has a plateau head after the elongation, so the representative height and diameter was found by the position were the first derivative went to 0.
6. The control fluctuating membrane was produced by using the reported simulation tool with a rigidity value of ~100 pN nm and no ligands (the height value was kept free) that was allowed to fluctuate until the energy reduced to a minimal value.

**Modelling and simulations.** The simulations are based on a rectangular array, typically of a size of few microns. The array is consisted of square 10 nm patches. The simulated area is the same as the experiment area. In order to achieve more realistic simulation results, we used a hybrid model where the locations of the TCR molecules are taken from our super-resolution experimental data. We used periodic boundary conditions (particles that exit on one side appear on the opposite side). The initial height ($z$) of the membrane patches is set to 50 nm except patches that contain a TCR where $z$ is set to 13 nm. The $z$ value of each patch is changing randomly at every iteration by $\Delta z$ that has a normal distribution with $\sigma = 1$ nm, and according to the Metropolis criterion.

We simulated four different types of proteins, as follows. TCR proteins behave as binding proteins to immobile ligands on a coverslip. The number and ($XY$) location of TCR molecules are taken from the experiments data. The $z$ coordinates of the TCRs are set to 13 nm at all times. CD45 proteins act as repulsive springs with the ability to bind to anti-CD45 ligands. The number of the simulated CD45 is constant throughout the simulation and is the mean of the experimental CD45 numbers between 10 and 60 s. Other glycoproteins (GPs), namely CD43 and CD148, are identical to the CD45 but do not bind to ligands. The total number of these GPs is twice the number of CD45 molecules. $XY$ locations of the CD45 molecules are simulated. The dynamics of the CD45 and the other GPs is governed

by minimizing the energy of the system and the confinement to a certain maximal apparent separation (MAS) distance from the TCRs, as detailed below.

**Monte-Carlo simulations**. Simulation energetic: In the simulations we used the Hamiltonian $H = H_{int} + H_{el}$, to calculate the energetics of the overall interactions between the T-cell membrane and the coverslip (represented by the term $H_{int}$) and the elasticity of the T cell membrane (represented by the term $H_{el}$). The interaction part, $H_{int}$, is defined as:

$$H_{int} = \sum_i \left( \delta_{1,CD45_i} \delta_{1,\alpha CD45_i} \right) V_{CD45-\alpha CD45}(z_i) + \delta_{1,CD45_i} V_{CD45}(z_i) \quad (16)$$

where,

$$\delta_{1,A_i} = \begin{cases} 1, & \text{if a protein of type A exists in patch } i \\ 0, & \text{otherwise} \end{cases} \quad (17)$$

A coverslip patch can contain only one ligand at a time. Likewise, a membrane patch can contain only one molecule at a time. In all the simulations TCRs are forced to be at $z = 13$ nm with no dependence on energy. The interaction potential of the CD45 with anti-CD45, $V_{CD45-\alpha CD45}$, is defined as:

$$V_{CD45-\alpha CD45}(z_i) = \begin{cases} U_{CD45-\alpha CD45}, & l_{CD45-\alpha CD45} - 5nm < z_i < l_{CD45-\alpha CD45} + 5nm \\ 0, & \text{elsewhere} \end{cases} \quad (18)$$

where $U_{CD45-\alpha CD45}$ is the interaction strength of a CD45 and anti-CD45, $l_{CD45-\alpha CD45}$ is the length of an engaged CD45-anti-CD45 (=35 nm). $z_i$ is the inter membrane-coverslip distance at patch $i$. The width of the square well potential was 10 nm and its depth was: $U_{CD45-\alpha CD45} = -10 \, K_B T$. The repulsion potential of the CD45 is defined as:

$$V_{CD45}(z_i) = \begin{cases} k_{CD45}(z_i - l_{CD45})^2, & z_i < l_{CD45} \\ 0 & z_i > l_{CD45} \end{cases} \quad (19)$$

where $k_{CD45} = 0.025 \, K_B T \, nm^{-2}$, is the compressional stiffness of the CD45 molecules and $l_{CD45}$, the length of the uncompressed CD45 molecules, is 50 nm. The elastic part of the Hamiltonian, $H_{el}$, is defined as:

$$H_{el} = \sum_i \frac{\kappa}{2a^2} (\Delta_d z_i)^2, \quad (20)$$

where $\kappa = \kappa_1 \cdot \kappa_2 / (\kappa_1 + \kappa_2)$, is the general effective bending rigidity of two membranes. In this case, the bending rigidity is effectively $\kappa \approx \kappa_1$, since $\kappa_2 \gg \kappa_1$ and is simulated at different values. The lattice constant, $a$, is 10 nm and $\Delta_d z_i = z_{i1} + z_{i2} + z_{i3} + z_{i4} - 4z_i$, (where $i1, i2, i3, i4$ are the indices of the four nearest neighbours of patch $i$).

Simulations dynamics: dynamics, time-step, accepting/rejecting attempts: The simulation propagates in time by iterations of 0.01 s. TCRs numbers and locations are updated every 2.5 s, (the experiment frame rate), from the experiment super-resolution data. In every iteration the CD45, and other GPs embedded in the plasma-membrane of the T cell, attempt to hop to one of the neighbouring patches. The CD45 and the other GPs have the same diffusion constant, $D_{CD45}$. The hopping attempts of the proteins are accepted or rejected according to the following rules:

1. The target patch is not occupied.
2. The probability of acceptance is according to Metropolis criterion:
   P(old state → free) = {1, $\Delta E < 0$, exp($-\Delta E$), $\Delta E > 0$} at old patch
   P(free → new state) = {1, $\Delta E < 0$, exp($-\Delta E$), $\Delta E > 0$} at new patch
   P(attempt accepted) = P(old state → free) × P(free → new state)
3. If more than one protein attempted to hop to the same patch, the protein with the highest energy gain would hop.
4. The height, $z$, of each patch of the membrane is changed randomly by $\Delta z$, that has a normal distribution with $\sigma = 1$ nm and according to Metropolis criterion. The value of $\sigma$ is set by receiving 40–50% of acceptance of the membrane attempts[33].

Simulations cases: We ran simulations for the two types of ligands used in the experiments, anti-CD3 and anti-CD45. In the anti-CD3 simulations we did not use ligands but forced the TCRs to be at $z = 13$ nm. In the anti-CD45 simulations we used anti-CD45 ligands and forced the TCRs to be at $z = 13$ nm. We scanned three different parameters: ligand density (0, 100, 300, 600, 1000, 3000 and 10,000 μm$^{-2}$), membrane rigidity (3, 10, 25, 100 [$K_B T$]) and 'maximal apparent separation' (MAS) (100, 150, 200, 250, 300 and 500 nm) of the segregation between TCRs and CD45 molecules. In the simulations we set limits on the locations of GPs (CD45, CD43 and CD148) so that GPs that are inside the area defined by the MAS are not allowed to jump out of it, while GPs that are outside this area are allowed to enter it.

The calculation of second-order statistics (i.e. pair correlation functions) and depletion zones for simulated data was carried out in the same manner as for the experimental part (see Methods, Statistics).

The simulations included 10$k$ steps of 250 × 250 to 450 × 500 pixels of 10 nm each and took 0.5 h (~100 s in cell time) each, using a PC (i7quad processor).

**Comparing simulations results to experimental results**. The comparison of the experimental and simulations results are divided to two parts. 1. Comparing the experimental and simulated images (Fig. 4d, e). TCR locations are taken from the experiment, while CD45 and other GPs can diffuse and interact. The overall appearance and evolution of the CD45 pattern in relation to the TCRs can be directly compared between the experiment and the simulations at different time-points and under various experimental conditions.

2. Comparing the depletion zones (TCR–CD45 separation distances) calculated from $g_{12}(r)$ curves (Fig. 5). To quantify differences between simulations and experiments, we compared the depletion zones calculated from the experiments and those found in the simulations (Fig. 5a, c). Before such a comparison can be made, we smoothed the experimental curves to match the temporal resolution of the simulations. Next, we can calculate differences between the depletion zones over time due to the experiments and the simulations (Fig. 5a, c; compare coloured curves to black curve). Such differences are described through the RMSD errors between the experimental and simulated curves (Fig. 5b, d). Varying physical parameters in the simulations, we next map the RMSDs for each condition (Fig. 5b, d). The lowest (blue) values mark the most predictive parameters, thus, the values that best describe the cells and molecules under study. We highlight these values with orange frames. Red frames in Fig. 5b, d, mark the parameters for which temporal curves are shown in Fig. 5a, c.

**Data availability**. The authors declare that the data supporting the findings of this study are available within the article and its supplementary information files, or are available upon reasonable requests to the authors.

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

## Acknowledgements

This research was supported by Grant no. 321993 from the Marie Skłodowska-Curie actions of the European Commission, the Lejwa Fund, and Grants no. 1417/13 and no. 1937/13 from the Israeli Science Foundation.

## Author contributions

E.S., M.R. supervised research; E.S., Y.R. designed research; Y.R. and J.S. developed reagents; Y.R. performed research; Y.N.-O. developed and performed simulations; Y.R. analysed the data; E.S., J.S. and Y.R. wrote the paper; E.S., J.S., Y.R., Y.N.-O. and M.R. commented on paper.

## Additional information

**Competing interests:** The authors declare no competing financial interests.

