## [Peer Review File · Nature Communications]

Reviewers' comments:

Reviewer #1 (PALM, SMLM, dSTORM, imaging)(Remarks to the Author):

Here, the Sherman group use multi-colour single-molecule super-resolution to shed light on the kinetic segregation model of T cell activation. This is quite nice work and I am supportive of the biology. However, the methodology is used in a way as though it were "routine" and there are not adequate controls for what is quite a novel methodology. While extensive revision is required, I do think this paper would become acceptable for publication:

The major limitation is that adequate controls are not in place for the 2-colour PALM/STORM combination. 1. Such imaging is subject to chromatic aberration. 2 spectral bleed through between the channels and 3. There may be issues with size exclusion when using fluorescent proteins and antibodies. There required controls are:

1. positive controls showing colocalization when the same protein is tagged by a FP and immunostained
2. That the same experimental results are obtained when the PALM/STORM labelling is reversed between the two proteins of interest
3. That the 2 colour imaging results in nothing in one channel when the experiments are conducted with only one of the labelling protocols.
4. Negative controls showing that two completely unconnected proteins show zero colocalization.

There is also a question of why the authors did not perform 3D imaging and analysis. This would certainly clarify much uncertainty about trying to interpret 3D systems using 2D data projections and would add considerable impact. 3D SMLM is now relatively routine.

There is also insufficient information to determine what effect multiple blinking has on this analysis. While it is mentioned in the SI, there are no values for the correction settings and it is not explained how any settings were determined. Moreover, it is known that correction for multiple blinking can't really be achieved for dSTORM. This therefore needs to be investigated further possibly through simulation or through repeating experiments with a dye that has very different blinking properties and showing that experimental results are unaffected.

It is also not clear what effect the localisation precision has, which the authors could test by repeating analysis with different uncertainty/photon filters on the localisations.

It is a bit of a limitation that all the results are performed in Jurkat Cells. Primary cells would give considerable more impact. I suggest that the key results are performed in primary human cells, or at least that basic control experiments are conducted to show basic statistics are valid from Jurkat cells. Drift correction of the data is not mentioned.

The use of the term "phase separation" in the introduction is a bit unclear. Can the authors clarify what they mean by this?

In general, the referencing is somewhat sparse and should probably include more of the literature on multi-colour PALM/STORM, co-localisation analysis of this kind of data, properties of the dyes/proteins etc.

Reviewer #2 (immunological synapse, TCR signaling)(Remarks to the Author):

This is a timely topic based on recent interest in the role of small projections in driving TCR signaling in response to pMHC may arise from the localization of TCR on F-actin based "microvilli" or "filopodia". This has been a long-standing notion, but only with TIRF based super-resolution localization microscopy and lattice light sheet microscopy has it been possible to resolve the steady state

structures and study them in more detail. Many questions are still unanswered. CD45 is partially excluded from TCR microclusters. But conventional resolution may miss the sub-diffraction segregation and there is also an issue that CD45 is needed to maintain activity of Lck. The paper covers a number of aspects from live localization microscopy on CD45 and TCR distribution using an unusual live hybrid PALM/dSTORM approach and AFM measurements on the T cell surface projections. These are quite diverse measurements on structures that maybe related functionally, but this is not entirely clear as the structures that are involved in probing surfaces may undergo some changes at the T cell contacts a surface. This leads to some diffuseness and also a surprising lack of rigor with regard to the labeling strategy for CD45, which on the face of it seems to be done in the worse possible way with a bivalent primary and bivalent secondary antibody at concentrations optimized to induce clustering. While the results are still intriguing there are many technical caveats that limit the utility of the data in its current form.

Major concerns

1. A major concern with this experiment is that the CD45 will be highly crosslinked with primary and secondary antibodies prior to the T cell coming into contact with the substrate and part of the experiment is based on using a anti-CD45 substrate, which would seem to use the same anti-CD45 antibody. How is the anti-CD45 on the substrate binding to the CD45 on the T cell surface if the anti-CD45 and the secondary antibody are engaging it. This is a very peculiar design and seems inferior to earlier efforts using anti-CD45 Fab, where the detection reagent had plausibly minimal impact on the bulk and the surface clustering of the CD45. The conditions used will certainly cluster the CD45 and should also induce its patching and capping over minutes.
2. Figure 1E shows Ca²⁺ results. I would like to verify that these cells are labeled with the anti-CD45 primary and secondary antibodies exactly as for the imaging experiments and the methods used to obtain this data should be specified- is this Fluo-4 or Fura-2, etc. Its not clear what is shown.
3. Its not clear how the videos were made. Is each frame from 0-2, 2-4, 4-6 ... seconds or 0-2, 0-4, 0-6 ... seconds? If it is the first case than it will look that they move because the following frames will not have the first frames, but if it is the second than they should be always there. I think they did it based on a first case. How do the molecules move on a fixed antibody substrates. Can it be assumed that both the PALM and dStorm approaches are bleaching fluorophores over time so that the location of the first TCR or CD45 to engage the substrate become invisible, but are still present and could be functional? This is a major problem as PALM should have this behavior and it means that localizations from all times would need to be included to construct images. Its not as clear if this should be true for dSTORM, although the compromise conditions needed to keep the cells alive may be more likely to lead to bleaching.
4. Another issue is that they TCR cluster sizes are massive, 500 nm (Fig2C). I have also concern about the labeling of the CD45 that I explain below regarding some figures. They also use 10mg/ml antibody to coat but I think its just a mistake in materials and methods and its 10microg/ml.
5. In Fig2A the image is not rendered based on the intensity, but as fixed dots. Its more typical to weight the dots in some way based on intensity/localization precision.
6. In Fig6: TCR (green) is CD3-Dronpa and it should colocalized with pCD3 (blue) but there is much more blue which does not colocalise at all and ist not clean how this can be unless there is some bleaching process that is not correctly compensated. Is this because of unlabeled endogenous CD3 that must then be preferentially utilized in favor to the dronpa labeled counterpart. It this is the case then it is not fair to make conclusions about the distances, because it can just happen that the one that is colocalising with pCD3 is invisible (unlabeled). They should use anti-TCR antibody (for example, IP26 clone for human TCR or H57 for mouse) to see whether all TCR has CD3-dronpa as positive control.
7. The authors are missing a positive control to show that how well they can achieve colocalization. On the Nikon system, when you use beads and you localise them, there is approximately 15-20 nm lateral

correction needed, due to chromatic aberration. Was this done?

8. FigS5 panel A: The amount of pCD3 molecules is much more as CD3, how can this be? This may arise from the method used to merged the data and why whatever approached was chosen needs to be explained. Also the amount of CD45 is less when stimulated with anti-CD45, is this because they use antibody that is blocking the binding side for the antibody they use to label CD45?

9. In one instance they discuss having 0.002 molecules per μm^2 ? But how could such sparse labeling be interpreted?

10. The measurements on the microvilli are interesting and the force-distance relationship in the pN range fits with values for TCR catch bond with around 10 pN over 10 nm being a relevant force. The only caveat is that these are projections on the free surface and its not clear if these are relevant for triggering. Is there a way to test this using a similar system with anti-CD3 beads as probe to confirm function of the projecting with the mechanical properties measured?

Minor concerns-

1. The images in Figure 1A are based on the anti-CD3 substrates developed by Bunnell et al in 2002 and that may be the most appropriate reference for microcluster formation. The Campi et al JEM 2005 deals more explicitly with f-actin and the timing of Calcium in relation to initial TCR microclusters so might be more appropriate than Yokosuka et al in this setting. So this reference went beyond the data in the earlier work from Bunnell et al and should also be cited, along with Yokosuka et al and the cited Varma et al paper that looked more at the requirement for F-actin in maintaining the TCR microclusters.

Reviewer #3 (TCR signaling, transcriptomic)(Remarks to the Author):

Using live-cell multi-color single-molecule-localization-microscopy, the authors have analyzed in a time-resolved manner the topological relationships existing between the TCR and CD45 in cellular structure identified as microvilli by using AFM. They made the paradoxical observation that TCR triggering - as documented by CD3zeta phosphorylation - negatively correlated with TCR-CD45 separation. This led them to revise the kinetic-segregation model and to suggest that in engaged microvilli the TCRs should be segregated, yet not removed too far, from CD45 for optimal activation. A major achievement of this study consists in analyzing the kinetic segregation of the TCR and CD45 with an unprecedented spatial resolution and to propose a solution to a paradox emphasized in a recent review by Weiss and Chakraborty (Nature Immunology 15, 798).

Main concern

As emphasized by recent, highly visible studies (doi/10.1073/pnas.1605399113 and dx.doi.org/10.1126/science.aal3118), the analysis of T cell microvilli and of their role in early T cell activation has to be performed at very early time points of activation since following activation, T cells rapidly 'spread' on the APC or on the activating surface which dramatically changed their morphology and the TCR distribution. The authors are aware of that vital issue since by discussing a former study (lane 56) they mention 'such contacts seems too late to influence early T cell activation'. Therefore, it is surprising to see that in the present study they used T cells that were fixed after 4 min of spreading on an αCD3 coated surface and also focused on microvilli that are located on the opposite face of the T-APC-interface ('at the apical side of live cells, as they adhered on coverslips'). Considering that T cells take decision on sub-minute time scale and through TCR located at the T-APC interface (not the apical side), the authors should explain how do they cope with the two experimental limitations outlined above to reach conclusions on the kinetic-segregation model, a model that concerns events that occur 'within seconds from TCR activation in engaged microvilli'.

Specific comments.

Lane 133 : Using the maximal and minimal values of BPCFS after performing 19 Monte-Carlo

simulations is certainly a very unreliable and questionable procedure. The authors should give at least a reference supporting this procedure and discuss this point.

Lane 173 : For many readers, it will be difficult to understand whether Kalman filtering is warranted and how this might provide the ability to retain the position of molecules.

Lane 383 : The KS between the TCR and CD45 occurred on non-stimulating coverslips has already observed by Chang et al. (Nature Immunology 17, 574).

Lane 209 : How can the AFM use high force mapping and effectively exert no spatial force on the the cell ? Along that line, it is certainly important for the reader to understand how contact is defined.

Lanes 215-218 : E dimension should be pn/nm^2 rather than pn.nm

Lane 221 : As described by Sage et al. (J. Immunol. 188:3686, 2012), the length of a T cell microvilli may be more than 400 nm.

Reviewer #4 (TCR signaling, costimulation)(Remarks to the Author):

The manuscript by Yair Razvag and co-authors gives novel insights into the molecular mechanisms of early T cell activation using super resolution imaging employing life cell multicolor signal molecule localization microscopy (SMLM) and atomic force microscopy. These technologies allow to determine the dynamic separation between T cell receptor (TCR) and CD45 phosphatases during cell contact upon TCR non-activation or TCR triggering. In addition, computational tools using physical modelling and simulation of respective image cell interfaces allowed the characterization of the dynamics of TCR-CD45 interaction, an approach, which could be further employed for analysis of other membrane proteins in a different context. These technologies improved the understanding of the TCR activation by identifying early contacts with microvilli under different TCR modulating conditions leading to a modulation of the kinetic segregation model. Despite this interesting topic the authors should address a number of issues:

- The results should be shown in another T cell line or primary T cells in order to generalize the results obtained.
- Does there exist differences in the segregation of TCR and CD45 in early T cell activation under physiologic and pathophysiologic conditions?
- There exists no information about how often the different experiments were performed.
- Information about time (kinetics) of TCR triggering/stimulation should be given in the manuscript text or figure legends and not only in the figures.
- Do there exist any differences between dynamic alterations during the synapse formation under different TCR stimulation.
- The segregation of TCR and CD45 using TCRs with different affinities should be analysed.
- The rational why cover slips were coded with α -CD11a, α -CD43 and α -CD148. What antibody coating served as negative controls for TCR triggering.

Reviewers' comments:

We would like to thank the reviewers for their detailed comments on our manuscript. Our point-by point address to the comments is marked below in brown. We have now extensively modified the manuscript and feel that these corrections have much improved the manuscript and its readability, towards possible publication.

Please note that in our revised manuscript, the order of multiple supplemental figures have changed. Our answers below always refer to the new figure location, wherever a figure is mentioned. Similarly, since multiple references have been added, the mentioned references below match the updated reference list. Specific textual changes are marked in red here, and in brown in the text of the revised manuscript.

Reviewer #1 (PALM, SMLM, dSTORM, imaging)(Remarks to the Author):

Here, the Sherman group use multi-colour single-molecule super-resolution to shed light on the kinetic segregation model of T cell activation. This is quite nice work and I am supportive of the biology. However, the methodology is used in a way as though it were "routine" and there are not adequate controls for what is quite a novel methodology. While extensive revision is required, I do think this paper would become acceptable for publication:

We much appreciate the support of the reviewer in publication of the paper. We have now added the required controls, and provided more details and references to better clarify and support our methodologies.

The major limitation is that adequate controls are not in place for the 2-colour PALM/STORM combination. 1. Such imaging is subject to chromatic aberration. 2 spectral bleed through between the channels and 3. There may be issues with size exclusion when using fluorescent proteins and antibodies.

We have now added multiple control experiment to address the concerns regarding our 2-colour PALM/dSTORM imaging approach.

There required controls are:

1. positive controls showing colocalization when the same protein is tagged by a FP and immunostained

To address this point, we have imaged the TCR via FP-tagging (PALM) and immunostaining (dSTORM). Specifically, we imaged cells stably expressing TCR ζ -Dronpa, while TCR molecules were stained using an antiTCR α/β primary antibody and a secondary stained with Alexa647. The cells were fixed after 4 min and imaged by PALM/dSTORM in TIRF mode. The data is shown in Supplementary Fig. 2C. The data shows high extent of colocalization of molecules in the two imaging channels (Dronpa in green and Alexa647 in red). As expected for such staining, the standardized bivariate PCF (blue line in Supplementary Fig. 2D) closely follows the model of random labelling (straight black lines around 0 in that panel).

2. That the same experimental results are obtained when the PALM/STORM labelling is reversed between the two proteins of interest

We have now recaptured the segregation between CD45 and the TCR via 2 color dSTORM using reversed colors, as follows (see Supplementary Fig. 2E,F). We stained CD45 with an α CD45 primary and a secondary antibody conjugated to Atto488 (shown in green), and TCR with α TCR ζ conjugated to Alexa647 (in red). The dSTORM images of fixed cells show the segregation of these molecules, as captured using the labeling approach throughout the manuscript, i.e. using TCR ζ -Dronpa and CD45-Alexa647 (Supplementary Fig. 2G,H). Indeed, the SBPCF statistics indicated a similar relative patterning of CD45 and the TCR at the PM of the cells using either the straight or the reversed coloring approaches.

3. That the 2 colour imaging results in nothing in one channel when the experiments are conducted with only one of the labelling protocols.

To exclude the possibility of cross-talk between the green and red channels in our PALM/dSTORM imaging, we imaged, using these two channels, cells that were labeled with either green molecules (stably expressing TCR ζ -Dronpa) or red molecules (α CD45-Alexa647). Imaging was done on fixed cells on α CD3 ϵ -coated coverslips. We show in Supplementary Fig. 1C that Dronpa does not show in the red channel. In the green channel, we note some low-level background throughout the imaging field. This uniform background is not due to cross-talk of Alexa647, since it did not show in the red channel. We conclude that no measurable cross-talk existed in our 2 color PALM/dSTORM imaging between the green and red imaging channels.

4. Negative controls showing that two completely unconnected proteins show zero colocalization.

To address this issue, we expressed Syntaxin1A-Dronpa in T cells and visualized it with CD45 stained with Alexa647 using PALM/dSTORM. Since Syntaxin1A is a transmembrane protein that originates from excitatory cells and is unrelated to T cells, we expect no interaction between this molecule and CD45. Indeed, our images (Supplementary Fig. 2A) shows effectively no colocalization of the two molecules. Our SBPCF statistics (Supplementary Fig. 2AB; blue line) indicates that these molecules follow the no-interaction model (red line) up to \sim 150nm. Deviation from this model is due to larger-scale PM patterning that affects both molecules. Such residual 'apparent' interaction is common to membrane proteins (see [21] Sajman, Sci.Rep. 2017;) and can be contrasted with our positive control using the 2 color staining of the same molecule (see Supplementary Fig. 2C,D).

There is also a question of why the authors did not perform 3D imaging and analysis. This would certainly clarify much uncertainty about trying to interpret 3D systems using 2D data projections and would add considerable impact. 3D SMLM is now relatively routine.

Following the reviewer's suggestion, we imaged using 3D PALM/dSTORM cells expressing TCR ζ -Dronpa and CD45, immunostained with Alexa647. The cells were fixed after 4 min from dropping onto α CD3 ϵ -coated coverslips. The new data is shown in Fig. 3E-H and in Supplementary Fig. 6C. As predicted from our 2D imaging and models, we find that TCR resides closer to the coverslip than CD45 (Fig. 3F, side view). Quantifying

this height separation in close contacts resulted in a consistent height difference of 60-70nm between these molecules. Statistics from multiple cells gave similar results (Supplementary Fig. 6C). We further discuss these new results, as follows:

“Our identification first contacts occurring at the tips of microvilli, and their characterization using AFM at the apical membrane of T cells, led us to visualize the 3D patterning of the segregation between the TCR and CD45 at the close contacts with the adhering part of the PM. For that, we turned to 3D PALM/dSTORM of TCR ζ -Dronpa and CD45 stained with Alexa647 in cells adhering on α CD3 ϵ -coated coverslips that were fixed after 4min (Fig. 3E,F; see Supplementary Note 1 for further details). Strikingly, we observed that TCR ζ and CD45 segregated not only in 2D (Fig. 3F top view, 3G), but also in their height relative to the coverslip (Fig. 3F, side view). This segregation was \sim 67nm in the close contact, although the z-positioning of the molecules at the PM had a relatively wide distribution of \sim \pm 22nm. This z separation across whole contacts of multiple cells was similar (\sim 88nm; Supplementary Fig. 6C).”

Technical details on the 3D imaging are given in a new section of the SI (Methods). We further conclude in the discussion:

“Three-dimensional PALM/dSTORM imaging allowed us to resolve the segregation of TCR and CD45 in close contacts of the cells with TCR-stimulating coverslips. This imaging revealed that these molecules segregated both in the plane of the coverslip and perpendicular to it, indicating coincidence of the KS with PM height differences, as expected at microvilli tips and PM ruffles.”

There is also insufficient information to determine what effect multiple blinking has on this analysis. While it is mentioned in the SI, there are no values for the correction settings and it is not explained how any settings were determined. Moreover, it is known that correction for multiple blinking can't really be achieved for dSTORM. This therefore needs to be investigated further possibly through simulation or through repeating experiments with a dye that has very different blinking properties and showing that experimental results are unaffected.

We agree with the reviewer that fluorophore blinking can lead to multiple counting of the same emitter in single molecule localization microscopy (using either PALM or dSTORM). We also agree that such blinking is often very challenging for dSTORM. Still, our main goal in this study was to capture the kinetic segregation of TCR and CD45 (i.e. their dynamic nanoscale patterning) in live, spreading cells. Thus, our focus has been on very fast imaging of these molecules rather than their precise counting. As a result, the rendering of our live cell images included no grouping of emitters over space and time. In our fixed imaging, we employed grouping over space and time as detailed in Supplementary Note 2: "For fixed cells experiments, a temporal gap of \sim 50 msec and a distance threshold of 20 nm were applied for each fluorophore separately in order to minimize possible over-counting of molecules."

We also note that our added imaging experiments using reversed colors for TCR and CD45 (in Supplementary Fig. 2E-H) demonstrate the segregation of these molecules using different dyes, with different blinking statistics.

It is also not clear what effect the localisation precision has, which the authors could test by repeating analysis with different uncertainty/photon filters on the localisations.

The localization precision of the different fluorophores in our PALM/dSTORM imaging is presented in Supplementary Fig 1A. PALM imaging of Dronpa resulted in localization errors of ~30nm, while dSTORM imaging of Alexa647 results in localization errors of ~15nm. Notably, such errors are much smaller than the key feature that we describe in our study, namely the depletion zone between TCR and CD45 molecules. This feature has a dynamic size of 100-300nm. Moreover, the localization errors are isotropic and should be RMS summed. This would only slightly affect our analyses of this feature by its smearing of up to a few percent (2-6%).

It is a bit of a limitation that all the results are performed in Jurkat Cells. Primary cells would give considerable more impact. I suggest that the key results are performed in primary human cells, or at least that basic control experiments are conducted to show basic statistics are valid from Jurkat cells.

Following the reviewer's suggestion, we repeated our experiments by PALM/dSTORM imaging of TCR ζ -Dronpa and CD45 in live J76 CD8+ T cells (i.e. a second T cell line). The results are presented in Supplementary Fig. 5C,D, and described in the text as follows:

“We next studied whether KS occurred only for for a second T cell line of a different lineage. For that, we conducted live cell PALM/dSTORM imaging of TCR ζ -Dronpa and CD45 at the PM of J76 CD8+ cells (see details in Supplementary Note 1) on α CD3 ϵ -coated coverslips. Our imaging showed distinct KS between TCR and CD45 also in these cells, with comparable depletion distance and dynamics to our results for Jurkat E6.1 cells (Supplementary Fig. 5C,D).”

Combined with the rest of the results in our study, we conclude the following in the main text:

“We conclude that KS occurs in both CD4+ and CD8+ T cells. It requires efficient spreading of the cells on coverslips, such as on α CD3 ϵ (UCHT1) and α CD11a. It also occurs on coverslips that do not stimulate the TCR well (either coated with α CD45 or lower concentrations of UCHT1), yet with a significantly lower extent and slower dynamics.” Thus, the kinetic segregation we found is common to multiple species of T cells.

Drift correction of the data is not mentioned.

We conducted drift correction for fixed cell imaging, as detailed in the SI. Still, we avoided such correction for live cell imaging, as explained in the added sentence to the section on Materials and Methods: **“Drift correction was applied for fixed-cell imaging, but not for live-cell imaging due to the fast rate and overall short duration of our imaging (2sec per an effective PALM/dSTORM frame).”**

The use of the term “phase separation” in the introduction is a bit unclear. Can the authors clarify what they mean by this?

We used the term ‘phase separation’ to describe the ‘physical separation’ of TCR and CD45 at the PM of T cells. However, ‘phase separation’ is a ‘reserved’ term in physics

and in chemistry, and is often used to describe pattern formation in spinodal fluids. Thus we have decided to replace this term with the terms ‘physical separation’ or ‘kinetic segregation’ of molecules throughout the text.

In general, the referencing is somewhat sparse and should probably include more of the literature on multi-colour PALM/STORM, co-localisation analysis of this kind of data, properties of the dyes/proteins etc.

We have now added multiple references to single molecule localization microscopy [(F)PALM and (d)STORM], including references to such imaging in multiple colors ([33]-[36] and in 3D [31], regarding possible related artefacts in co-localization and molecular counting [37], and regarding the photophysics of Dronpa [23].

Reviewer #2 (immunological synapse, TCR signaling)(Remarks to the Author):

This is a timely topic based on recent interest in the role of small projections in driving TCR signaling in response to pMHC may arise from the localization of TCR on F-actin based “microvilli” or “filopodia”. This has been a long-standing notion, but only with TIRF based super-resolution localization microscopy and lattice light sheet microscopy has it been possible to resolve the steady state structures and study them in more detail. Many questions are still unanswered. CD45 is partially excluded from TCR microclusters. But conventional resolution may miss the sub-diffraction segregation and there is also an issue that CD45 is needed to maintain activity of Lck. The paper covers a number of aspects from live localization microscopy on CD45 and TCR distribution using an unusual live hybrid PALM/dSTORM approach and AFM measurements on the T cell surface projections. These are quite diverse measurements on structures that maybe related functionally, but this is not entirely clear as the structures that are involved in probing surfaces may undergo some changes at the T cell contacts a surface. This leads to some diffuseness and also a surprising lack of rigor with regard to the labeling strategy for CD45, which on the face of it seems to be done in the worse possible way with a bivalent primary and bivalent secondary antibody at concentrations optimized to induce clustering. While the results are still intriguing there are many technical caveats that limit the utility of the data in its currentl form.

We thank the reviewer for his positive view on the timeliness of our study and the novelty of our techniques and findings. We also much appreciate his technical comments, which we have addressed via multiple experiments and rewriting.

Major concerns

1. A major concern with this experiment is that the CD45 will be highly crosslinked with primary and secondary antibodies prior to the T cell coming into contact with the substrate and part of the experiment is based on using a anti-CD45 substrate, which would seem to use the same anti-CD45 antibody. How is the anti-CD45 on the substrate binding to the CD45 on the T cell surface if the anti-CD45 and the secondary antibody are engaging it. This is a very peculiar design and seems inferior to earlier efforts using anti-CD45 Fab, where the detection reagent had plausibly minimal impact on the bulk and the surface clustering of the CD45. The conditions

used will certainly cluster the CD45 and should also induce its patching and capping over minutes.

The reviewer raises two concerns related to our use of antiCD45 antibodies; first, our use of these antibodies in immunostaining of CD45 at the PM of cells; and second, the use of these antibodies in our imaging assays for generating antibody-coated coverslips.

In response to the first concern regarding the use of α CD45 for immunostaining:

We agree with the reviewer that α CD45 antibodies may induce clustering at the PM of cells, prior to their engagement of the coverslips. Unfortunately, CD45 FABs are not commercial and we could not obtain them for the current study. Thus, taking the advice of the reviewer to minimize such a possible effect, we repeated the live cell imaging experiments using TCR ζ -Dronpa and a primary α CD45 antibody directly conjugated to Alexa647. The cells were dropped on α CD3 ϵ -coated coverslips. The results are presented in Supplementary Fig. 5A and show distinct kinetic segregation of TCR and CD45. Quantification of the depletion zone and its dynamics (Supplementary Fig. 5B) showed similar results between immunostaining with a primary (magenta line) and immunostaining with a primary and a secondary (as typically performed in the study; black line). These new results support the validity of our results regarding the KS of TCR and CD45 and its dynamics.

To further test for possible effects of CD45 staining on the TCR-CD45 patterning we observed, we imaged cells that were either stained before dropping on the coverslips (as typically performed in our study) or after fixation (which was done 4 min after the dropping of the cells on the coverslips). The first approach is compatible with live cell imaging, while the second approach rules out the involvement of CD45 immunostaining with its patterning at the PM, relative to TCR. The cells were dropped on α CD11a-coated coverslips for both immunostaining approaches. Our results, presented below, show similar patterning and SBPCF curves for the two immunostaining approaches (panels A,B for immunostaining before fixation, and panels C,F for immunostaining after fixation).

To address the second point regarding our use of α CD45 antibodies for coating coverslips, we would like to point out the following:

First, our measurements of the kinetic segregation between TCR and CD45 were conducted on multiple types of antibody-coated coverslips, including α CD3 ϵ , α CD11 or α CD45. Under all conditions, we found striking patterns of kinetic segregation between the TCR and CD45 and analyzed them using various statistical measures. Thus, our conclusions on the kinetic segregation and its inducing mechanisms are not restricted to the results on α CD45-coated coverslips. Moreover, the CD45/TCR patterns that we detected using live cell microscopy show that CD45 patterns are dynamic and are not limited by the adhesion of cells to the coverslips, regardless of the antibody-coating used (see for instance Fig. 2A).

Second, and importantly, we quantified the clustering sizes of CD45 on the different antibody-coated coverslips, and found no effect of these antibodies on the maximal CD45 cluster sizes and their dynamics (see Fig. 2D).

Third, our immunostaining of CD45 at the PM of cells prior to their dropping on coverslips may not saturate all of the available CD45 molecules. If so, there will be enough CD45 molecules available for binding by the α CD45 antibodies on the coverslip. Moreover, our use of poly-L-lysine also promotes nonspecific cell adhesion. Thus, the major goal of the α CD45-coated coverslips is achieved in getting cell adhesion and spreading without directly engaging and stimulating the TCR.

2. Figure 1E shows Ca²⁺ results. I would like to verify that these cells are labeled with the anti-CD45 primary and secondary antibodies exactly as for the imaging experiments and the methods used to obtain this data should be specified- is this Fluo-4 or Fura-2, etc. Its not clear what is shown.

Our Ca^{2+} data was meant to verify that the cells were activated on $\alpha\text{CD3}\epsilon$ -coated coverslips, but not on αCD11a or αCD45 -coated coverslips. Unfortunately, Dronpa emission significantly overlaps with Fluo-4AM, while our system is not suited for Fura-2 imaging. This prevented our imaging of cells with the same cells as in the imaging experiments. We further point out that our study included multiple other measurements involving T cell activation, including cell spreading (Fig. 1B-D), TCR clustering (Fig. 2A,C) and TCR phosphorylation (Fig. 6). All of these assays showed robust T cell activation on the $\alpha\text{CD3}\epsilon$ -coated coverslips. We have now added a dedicated section to Supplementary Note 1, describing the Ca^{2+} measurements in more detail.

3. Its not clear how the videos were made. Is each frame from 0-2, 2-4, 4-6 ... seconds or 0-2, 0-4, 0-6 ... seconds? If it is the first case than it will look that they move because the following frames will not have the first frames, but if it is the second than they should be always there. I think they did it based on a first case.

Indeed, our live cell imaging was 0-2, 2-4, 4-6, etc. We now clarify this in the Materials and Methods.

How do the molecules move on a fixed antibody substrates. Can it be assumed that both the PALM and dStorm approaches are bleaching fluorophores over time so that the location of the first TCR or CD45 to engage the substrate become invisible, but are still present and could be functional? This is a major problem as PALM should have this behavior and it means that localizations from all times would need to be included to construct images. Its not as clear if this should be true for dSTORM, although the compromise conditions needed to keep the cells alive may be more likely to lead to bleaching.

In our PALM/dSTORM imaging we show clear dynamic patterning of both TCR ζ and CD45. This is especially evident in our movies that show cell spreading from initial contacts (Fig. 2 and Supplementary Movies 1,2). As mentioned before (in response to the first concern), our immunostaining of CD45 at the PM of cells prior to their dropping on coverslips may not saturate all of the available CD45 molecules. If so, there will be enough mobile CD45 molecules appearing at the cell footprint. Moreover, our use of poly-L-lysine also promotes nonspecific cell adherence that further allows for CD45 mobility on αCD45 -coated coverslips.

We do observe fast photobleaching for Dronpa in PALM imaging, and less so for Alexa647 in dSTORM mode. To compensate for this fast bleaching of Dronpa, we employed Kalman filtering that carried some memory of the locations of emitters in previous frames (as detailed in Supplementary Note 2). Such filtering was not necessary for Alexa647, as these fluorophores underwent repeated blinking events and remained fluorescent for relatively longer imaging durations.

We now clarify these points in Supplementary Note 2, and in the main text: “Kalman filtering was not necessary for Alexa647, due to its robust emission for dSTORM [22]”.

4. Another issue is that they TCR cluster sizes are massive, 500 nm (Fig2C). I have also concern about the labeling of the CD45 that I explain below regarding some figures. They also

use 10mg/ml antibody to coat but I think its just a mistake in materials and methods and its 10microg/ml.

Regarding the antibody coating concentration - Indeed, the coating concentration using either α CD3 ϵ , α CD45 or α CD11a was 10 micrograms/ml. We have now corrected this concentration in the text, and thank the reviewer for pointing out this mistake.

Regarding to the quantification of TCR and CD45 clustering - We wanted to clarify that the reported values are for the largest cluster sizes of these molecules rather than their average. These reported measures are found by the statistics that we employed, namely univariate pair correlation functions (PCFs). They are determined by the length-scale in which the PCF curves cross with a value of 1, which represents complete spatial randomness (CSR). Thus, these values should be interpreted as the largest scale of significant self-clustering of each molecule, and should not be directly compared with previously reported average cluster sizes. Using this measure, we aimed to show that the extent of self-clustering of CD45 and TCR does not change much upon cell spreading around the early contacts. Since our molecular counting is imprecise, we felt that this measure is more appropriate than describing average cluster sizes using clustering algorithms. To clarify this point, we have modified the labels in the relevant figure (Fig. 2C,D) and in its legend.

5. In Fig2A the image is not rendered based on the intensity, but as fixed dots. Its more typical to weight the dots in some way based on intensity/localization precision.

Dot representation of single emitted in SMLM images emphasizes sparsely detected emitters vs. clusters. For our live-cell imaging, we preferred to render the data in this way such that newly recruited molecules can be intuitively related to the apparent cell contact. We write in the section on Materials and Methods in the main text:

“SMLM movies (and individual frames) show individual molecules as dots. We preferred this representation over alternative representations (such as individual or summed Gaussians 15) since it better shows sparse and newly recruited molecules, and their position relative to the cell contact.”

6. In Fig6: TCR (green) is CD3-Dronpa and it should colocalized with pCD3 (blue) but there is much more blue which does not colocalise at all and ist not clean how this can be unless there is some bleaching process that is not correctly compensated. Is this because of unlabeled endogenous CD3 that must then be preferentially utilized in favor to the dronpa labeled counterpart. It this is the case then it is not fair to make conclusions about the distances, because it can just happen that the one that is colocalising with pCD3 is invisible (unlabeled). They should use anti-TCR antibody (for example, IP26 clone for human TCR or H57 for mouse) to see whether all TCR has CD3-dronpa as positive control.

As suggested by the reviewer, we imaged TCR ζ -Dronpa by PALM and TCR immunostained with the primary α TCR α/β (IP26) and a secondary conjugated to Alexa647 by dSTORM in the same cells. As expected, our imaging shows high colocalization of the two labels (Supplementary Fig. 2C). The SBPCF statistics (Supplementary Fig. 2D) shows that the mutual patterning of the molecules approximates the model of random labeling

(blue line vs. black line). Occasionally, we do observe non-uniform staining by the α TCR α/β antibody.

Moreover, we note that our cells include endogenous TCR ζ molecules without the Dronpa tag, which would give rise to some pTCR ζ staining without Dronpa emission (points in blue without green, in Fig 6A,B).

Nevertheless, our statistical analyses for the 3 color images in Fig.6 are insensitive to the non-uniform staining of TCRs by the α pTCR ζ antibody (in blue) and the endogenous population of TCR, since neither of them depend on the placement of CD45 (red) relative to TCR molecules (either blue or green).

7. The authors are missing a positive control to show that how well they can achieve colocalization. On the Nikon system, when you use beads and you localise them, there is approximately 15-20 nm lateral correction needed, due to chromatic aberration. Was this done? To demonstrate our ability to colocalize emitters in different colors, we now added images of Tetraspec beads in 3 imaging colors (Supplementary Fig. 1B). In our study, we use SBPCF statistics to check for interaction between point patterns (e.g. Supplementary Fig. 1B). In Supplementary Fig. 1C, the SBPCF statistics was applied for the green and red channels (as a representative example) in the data of Supplementary Fig. 1B. The statistics show that the data is consistent with the 95% confidence interval due to a random labeling model, which further indicates excellent registration of the red and green channels.

8. FigS5 panel A: The amount of pCD3 molecules is much more as CD3, how can this be? This may arise from the method used to merged the data and why whatever approached was chosen needs to be explained. Also the amount of CD45 is less when stimulated with anti-CD45, is this because they use antibody that is blocking the binding side for the antibody they use to label CD45?

The labeling efficiencies of TCRs using α pTCRs antibodies or by the genetically-encoded fluorescent protein Dronpa cannot be directly compared. Moreover, the counting by dSTORM typically suffers from multiple artifacts that may lead to both over and undercounting of the detected molecules. Thus, the protein levels in Fig. S5A cannot be directly compared. To clarify this point, we have added the following to the main text: **“Importantly, our PALM/dSTORM imaging aimed to resolve the KS between the TCR and CD45 and was not optimized for the absolute counting of these of these molecules. Such counting is imprecise, esp. for live cell imaging and for dSTORM [31] (see Supplementary Notes 1,2 for further details on SMLM imaging and analyses).”**

Regarding the relative amount of CD45 - Our assay included labeling of CD45 before dropping the cells on the antibody-coated coverslips. Thus, the α CD45 coating on coverslips should not interfere with CD45 labeling at the PM. Indeed, the relative abundance of CD45 molecules at the PM of activated cells was surprising and may be related to active recruitment of these molecules to the early contacts.

9. In one instance they discuss having 0.002 molecules per μm^2 ? But how could such sparse labeling be interpreted?

Indeed, we had a mistake with the units and we thank the reviewer for pointing out this mistake. We have now corrected the numbers in Fig. S5A (which should be around 200 molecules per μm^2).

10. The measurements on the microvilli are interesting and the force-distance relationship in the pN range fits with values for TCR catch bond with around 10 pN over 10 nm being a relevant force. The only caveat is that these are projections on the free surface and its not clear if these are relevant for triggering. Is there a way to test this using a similar system with anti-CD3 beads as probe to confirm function of the projecting with the mechanical properties measured?

The reviewer is raising an intriguing possibility of the involvement of TCR catch bonds in the applied forces. Unfortunately, our current setup excludes force measurements on the adhering side of the cell. Such measurements would largely exceed the scope of the current study, and we will consider the possibility of conducting them as a follow-up study.

Minor concerns-

1. The images in Figure 1A are based on the anti-CD3 substrates developed by Bunnell et al in 2002 and that may be the most appropriate reference for microcluster formation. The Campi et al JEM 2005 deals more explicitly with f-actin and the timing of Calcium in relation to initial TCR microclusters so might be more appropriate than Yokosuka et al in this setting. So this reference went beyond the data in the earlier work from Bunnell et al and should also be cited, along with Yokosuka et al and the cited Varma et al paper that looked more at the requirement for F-actin in maintaining the TCR microclusters.

We thank the reviewer for his comments of the appropriate references, and have modified our citations accordingly.

Reviewer #3 (TCR signaling, transcriptomic)(Remarks to the Author):

Using live-cell multi-color single-molecule-localization-microscopy, the authors have analyzed in a time-resolved manner the topological relationships existing between the TCR and CD45 in cellular structure identified as microvilli by using AFM. They made the paradoxical observation that TCR triggering - as documented by CD3zeta phosphorylation - negatively correlated with TCR-CD45 separation. This led them to revise the kinetic-segregation model and to suggest that in engaged microvilli the TCRs should be segregated, yet not removed too far, from CD45 for optimal activation. A major achievement of this study consists in analyzing the kinetic segregation of the TCR and CD45 with an unprecedented spatial resolution and to propose a solution to a paradox emphasized in a recent review by Weiss and Chakraborty (Nature Immunology 15, 798).

We much appreciate the reviewer's acknowledgement of the novelty of our imaging and the significance of the results in our study.

Main concern

As emphasized by recent, highly visible studies (doi/10.1073/pnas.1605399113 and dx.doi.org/10.1126/science.aal3118), the analysis of T cell microvilli and of their role in early T cell activation has to be performed at very early time points of activation since following activation, T cells rapidly 'spread' on the APC or on the activating surface which dramatically changed their morphology and the TCR distribution. The authors are aware of that vital issue since by discussing a former study (lane 56) they mention 'such contacts seems too late to influence early T cell activation'. Therefore, it is surprising to see that in the present study they used T cells that were fixed after 4 min of spreading on an α CD3 coated surface and also focused on microvilli that are located on the opposite face of the T-APC-interface ('at the apical side of live cells, as they adhered on coverslips'). Considering that T cells take decision on sub-minute time scale and through TCR located at the T-APC interface (not the apical side), the authors should explain how do they cope with the two experimental limitations outlined above to reach conclusions on the kinetic-segregation model, a model that concerns events that occur 'within seconds from TCR activation in engaged microvilli'.

We completely agree with the reviewer that TCR decision making is performed in sub-minute timescales and in early contacts. Indeed, characterizing these early timescales and small scale of contacts has posed a significant experimental challenge for 3-color PALM/dSTORM imaging. Our solution was to develop and implement fast, two-color single molecule localization microscopy of live cells, with spatial resolution down to ~20nm, and with an effective temporal resolution of 2-3sec. Indeed, most of our findings regarding the kinetic segregation of TCR and CD45 in early contacts rely on these measurements.

Imaging was done on fixed cells after 4 mins only for 3-color PALM/dSTORM imaging, to get robust spreading of cells with as many detected molecules possible needed for our statistics. Indeed, this time point captures a range of contacts, where very early contacts are only sparsely represented.

Taking the reviewer's suggestion, we have now added 3 color imaging of cells that were fixed after 1 min from dropping on α CD3 ϵ -coated coverslips. The results are presented in Supplementary Fig. 9G-I. We describe the measurements and results in the main text, as follows: "Our three-colour SMLM imaging and analyses of the organization of TCR ζ , pTCR ζ and CD45 required relatively wide cell footprints, so that enough molecules from each species could be detected. Nevertheless, KS of TCR and CD45 may lead to T cell activation with seconds from first contact. Thus, we repeated these experiments for cells that were fixed after 1 min from dropping onto α CD3-coated coverslips. Our results for this earlier time-point could also detect the enrichment of pTCR ζ in TCR ζ molecules proximal (yet segregated) from CD45, as indicated by cell images (Supplementary Fig. 9G) and statistics [Supplementary Fig. 9I (right); albeit being noisy due to the low counts of detected molecules (compare panels A,C with panels H,I in Supplementary Fig. 9)]."

Regarding our AFM measurements on the apical side of the cell - These measurements were aimed at: (1) Characterizing the microvilli size *before* cell engagement, since we hypothesize that early contacts of T cells is achieved via microvilli. This hypothesis is further supported by our SMLM imaging using a membrane dye, which shows similar initial footprints of the tips of microvilli at the PM in the

adhering side of the cell (Fig. 3I,J). (2) Measuring the PM rigidity using quantitative imaging (QI). Both goals have been achieved and are appropriate for conducting on the apical membrane of cells, as they adequately represent non-engaged parts of the plasma membrane. Unfortunately, such measurements on the adhering side of the cell cannot be performed using our AFM system, and would largely exceed the scope of the current study.

Specific comments.

Lane 133 : Using the maximal and minimal values of BPCFS after performing 19 Monte-Carlo simulations is certainly a very unreliable and questionable procedure. The authors should give at least a reference supporting this procedure and discuss this point.

Our approach for measuring the confidence limits of ‘null hypotheses’ using PCF (and SBPCF) statistics is based on a common methodology called ‘bootstrapping’. Through this approach, we simulate $n=19$ realizations of point-patterns according to a specific model, and then check whether the PCF statistics match or deviate from the highest and lowest statistics due to the simulated data. If the data lies within this range, the model can be accepted with a confidence of $n/(n+1) = 95\%$. If not, the model can be rejected with similar confidence. We, and others, have used this approach successfully and extensively before to distinguish underlying processes in point patterns (e.g. [19] Sherman et al, Methods, 2013). We have added a well cited reference ([20] Wiegand and Moloney, Oikos, 2004), on which we have previously relied, to further highlight the reliability and credibility of this approach.

Lane 173 : For many readers, it will be difficult to understand whether Kalman filtering is warranted and how this might provide the ability to retain the position of molecules.

In our PALM imaging we observe bleaching of Dronpa. To compensate for this bleaching, we employed Kalman filtering that carried some memory of the locations of emitters in previous frames (Kalman filtering is described in Supplementary Note 2.2). The effect of this filtering is demonstrated in Supplementary Fig. 4B, where the appearance of photobleached Dronpa molecules is carried from one frame to a following one through Kalman filtering.

Lane 383 : The KS between the TCR and CD45 occurred on non-stimulating coverslips has already observed by Chang et al. (Nature Immunology 17, 574).

We thank the reviewer for pointing this, and acknowledge that these findings are consistent with earlier findings by Chang et al (ref [11]).

Lane 209 : How can the AFM use high force mapping and effectively exert no spatial force on the the cell ? Along that line, it is certainly important for the reader to understand how contact is defined.

In our AFM measurements, the force curves were conducted on each pixel and the AFM tip was retracted after each measurement in individual pixels. In this way, effectively no lateral force is exerted on the surfaces. We have now added the text below to clarify this point and the definition of the contact in relation to the AFM measurements:

“In this technique, force curves are measured on individual pixels as follows. The place in which the force is measured by the AFM is first defined by the AFM tip (2nm radius) which is brought into contact with the PM until a certain predefined counter-force is reached. The AFM cantilever (and tip) are elevated after each force curve, such that effectively no spatial force is exerted on the cell surface (Supplementary Fig. 6A) [24]”.

Lanes 215-218 : E dimension should be pn/nm² rather than pn.nm

To be consistent with our simulations of the PM, and the simulations of others in the field, we transformed the results of the Young’s modulus (pn / nm²) to bending rigidity (pn x nm). The details are specified in the SI. We now clarify this point in the text:

“Note that we have translated Young’s modulus here to rigidity modulus, following Deserno 26 (see Supplementary Note 3), for consistency with modelling and simulations that we introduce below.”

Lane 221 : As described by Sage et al. (J. Immunol. 188:3686, 2012), the length of a T cell microvilli may be more than 400 nm.

We thank the reviewer for raising this point. Our measurements show that microvilli extend have a typical length of 100-200nm (Fig. 3B). Indeed, Sage et al report on invadosome-like protrusions (ILPs),that extend significantly longer from the PM of closely interacting cells. Such protrusions are likely different in nature from the microvilli that are engaged in first contacts (which we describe). We have now added a reference to Sage et al ([26]) and mentioned that “longer PM protrusions have been reported” by this study.

Reviewer #4 (TCR signaling, costimulation)(Remarks to the Author):

The manuscript by Yair Razvag and co-authors gives novel insights into the molecular mechanisms of early T cell activation using super resolution imaging employing life cell multicolor signal molecule localization microscopy (SMLM) and atomic force microscopy. These technologies allow to determine the dynamic separation between T cell receptor (TCR) and CD45 phosphatases during cell contact upon TCR non-activation or TCR triggering. In addition, computational tools using physical modelling and simulation of respective image cell interfaces allowed the characterization of the dynamics of TCR-CD45 interaction, an approach, which could be further employed for analysis of other membrane proteins in a different context. These technologies improved the understanding of the TCR activation by identifying early contacts with microvilli under different TCR modulating conditions leading to a modulation of the kinetic segregation model. Despite this interesting topic the authors should address a number of issues:

We appreciate the reviewer’s comments on the significance of our results and the diversity and applicability of our developed methods in this study.

- The results should be shown in another T cell line or primary T cells in order to generalize the results obtained.

This point has been raised also by the first reviewer, and we repeat below our answer to this comment:

Following the reviewer's suggestion, we repeated our experiments by PALM/dSTORM imaging of TCR ζ -Dronpa and CD45 in live J76 CD8+ T cells (i.e. a second T cell line). The results are presented in Supplementary Fig. 5C,D, and described in the text as follows:

"We next studied whether KS occurred only for a second T cell line of a different lineage. For that, we conducted live cell PALM/dSTORM imaging of TCR ζ -Dronpa and CD45 at the PM of J76 CD8+ cells (see details in Supplementary Note 1) on α CD3 ϵ -coated coverslips. Our imaging showed distinct KS between TCR and CD45 also in these cells, with comparable depletion distance and dynamics to our results for Jurkat E6.1 cells (Supplementary Fig. 5C,D)."

Combined with the rest of the results in our study, we conclude the following in the main text:

"We conclude that KS occurs in both CD4+ and CD8+ T cells. It requires efficient spreading of the cells on coverslips, such as on α CD3 ϵ (UCHT1) and α CD11a. It also occurs on coverslips that do not stimulate the TCR well (either coated with α CD45 or lower concentrations of UCHT1), yet with a significantly lower extent and slower dynamics." Thus, the kinetic segregation we found is common to multiple types of T cells.

- Does there exist differences in the segregation of TCR and CD45 in early T cell activation under physiologic and pathophysiologic conditions?

Our study has focused on measuring intact cells (being either fixed or live). It is an intriguing possibility that pathophysiological conditions may alter the kinetic segregation between the TCR and CD45 and the resultant TCR activation. However, such measurements exceed the scope of the current study and its findings, and would be considered in separate, follow-up study. We propose that our results could "...set the stage for studying the KS role in health and disease".

- There exists no information about how often the different experiments were performed.

We have now added the following to Supplementary Note 1:

"Experiments involving live and fixed cell imaging were repeated 2-4 times for each measurement. Analyses (Supplementary Note 2) included data from all repetitions."

- Information about time (kinetics) of TCR triggering/stimulation should be given in the manuscript text or figure legends and not only in the figures.

We now added the time for TCR triggering also in the figure legends.

- Do there exist any differences between dynamic alterations during the synapse formation under different TCR stimulation.

To address this point, we have measured the KS for live cells spreading on coverslips coated with a low concentration of α CD3 ϵ (UCHT1; 1 μ g/ml]. Under this condition, the cells failed to spread efficiently, and no kinetic segregation was observed between the TCR and CD45 (Supplementary Fig. 5E,F).

- The segregation of TCR and CD45 using TCRs with different affinities should be analysed.

The specificity of the TCRs in the Jurkat E6.1 cells under our study is unknown. For checking the TCR affinity effect on the observed KS, our experimental approach should be significantly expanded to use cells with TCR recognition of altered-peptides, presented on APCs or on glass-supported lipid bilayers. Such changes would constitute major changes in our experimental system and largely exceed the scope of the current study. Thus, we ask to leave this important and interesting research question, which goes beyond our current findings, to a future study.

- The rationale why cover slips were coated with α -CD11a, α -CD43 and α -CD148. What antibody coating served as negative controls for TCR triggering.

It is apparent that any engagement of T cells with a surface leads to some signaling events of the cells. Thus, we used coverslips first coated with poly-L-lysine and subsequently, with either α CD11a or α CD45 as negative controls for TCR triggering, as they do not directly bind the TCR. We do observe that these coatings do not trigger Ca^{2+} flux in the adhering cells (Fig. 1E), however we note that the α CD45 coating may cause limited phosphorylation of the TCR (Supplementary Fig. 9A).

REVIEWERS' COMMENTS:

Reviewer #1 (Remarks to the Author):

I think the authors have really done a very good job addressing the comments - they were quite extensive

I consider the manuscript acceptable for publication...

Reviewer #2 (Remarks to the Author):

The authors have done a number of controls that that were requested by the STORM expert and have justified some choices they have made regarding the biology. They have also performed the experiments using directly labelled anti-CD45 antibodies and explained how the the use of the anti-CD45 antibody on a substrate could work. The most compelling result is that TCR phosphorylation is greatest near to, but no overlapping with anti-CD45 clusters. This upholds expectations from CD45 KO mice that maintaining active Lck is likely a very important function and being too far from CD45 likely reduces TCR signaling. Therefore, I feel that the changes and improved the paper and I supports in publication. Caveats of the probes and specific substrates should be evident and well described.

Reviewer #3 (Remarks to the Author):

None

Reviewer #4 (Remarks to the Author):

The revised version of the manuscript by Razvag and co-authors has improved the paper a lot. The authors addressed all the major queries and integrated them into the manuscript and included a number of additional experiments and controls. Furthermore, they also included data from a second T cell line of a different lineage and further extended the information concerning the contact regions at microvilli tips and visualisation of the segregation of the TCR and CD45. The new results were also discussed thoroughly and more information was given concerning the PALM imaging data and their analyses in the method section.

Reviewers' comments:

We would like to thank the reviewers once again for their detailed comments on our manuscript, which have significantly improved our study. We also thank them for their kind support in publishing this work.

REVIEWERS' COMMENTS:

Reviewer #1 (Remarks to the Author):

I think the authors have really done a very good job addressing the comments - they were quite extensive

I consider the manuscript acceptable for publication...

Reviewer #2 (Remarks to the Author):

The authors have done a number of controls that that were requested by the STORM expert and have justified some choices they have made regarding the biology. They have also performed the experiments using directly labelled anti-CD45 antibodies and explained how the the use of the anti-CD45 antibody on a substrate could work. The most compelling result is that TCR phosphorylation is greatest near to, but no overlapping with anti-CD45 clusters. This upholds expectations from CD45 KO mice that maintaining active Lck is likely a very important function and being too far from CD45 likely reduces TCR signaling. Therefore, I feel that the changes and improved the paper and I supports in publication. Caveats of the probes and specific substrates should be evident and well described.

Reviewer #3 (Remarks to the Author):

None

Reviewer #4 (Remarks to the Author):

The revised version of the manuscript by Razvag and co-authors has improved the paper a lot. The authors addressed all the major queries and integrated them into the manuscript and included a number of additional experiments and controls. Furthermore, they also included data from a second T cell line of a different lineage and further extended the information concerning the contact regions at microvilli tips and visualisation of the segregation of the TCR and CD45. The new results were also discussed thoroughly and more information was given concerning the PALM imaging data and their analyses in the method section.